# Angiotensin II induces coordinated calcium bursts in aldosterone-producing adrenal rosettes

Nick A. Guagliardo[1], Peter M. Klein[1,2], Christina A. Gancayco[3], Adam Lu [1,2], Sining Leng[4],
Rany R. Makarem [1], Chelsea Cho[1], Craig G. Rusin[5], David T. Breault[4], Paula Q. Barrett [1] &
Mark P. Beenhakker[1,2 ✉]

Aldosterone-producing zona glomerulosa (zG) cells of the adrenal gland arrange in distinct multi-cellular rosettes that provide a structural framework for adrenal cortex morphogenesis and plasticity. Whether this cyto-architecture also plays functional roles in signaling remains unexplored. To determine if structure informs function, we generated mice with zG-specific expression of GCaMP3 and imaged zG cells within their native rosette structure. Here we demonstrate that within the rosette, angiotensin II evokes periodic $Ca_v3$-dependent calcium events that form bursts that are stereotypic in form. Our data reveal a critical role for angiotensin II in regulating burst occurrence, and a multifunctional role for the rosette structure in activity-prolongation and coordination. Combined our data define the calcium burst as the fundamental unit of zG layer activity evoked by angiotensin II and highlight a novel role for the rosette as a facilitator of cell communication.

[1] Departments of Pharmacology, Charlottesville, VA, USA. [2] Neuroscience Graduate Program, University of Virginia, Charlottesville, VA, USA. [3] Research Computing, University of Virginia, Charlottesville, VA, USA. [4] Division of Endocrinology, Boston Children's Hospital, Boston, MA, USA. [5] Department of Pediatrics-Cardiology, Baylor College of Medicine, and Harvard Stem Cell Institute, Cambridge, MA, USA. ✉email: mpb5y@virginia.edu

The aldosterone-producing zona glomerulosa (zG) layer of the adrenal gland is in a constant state of self-renewal[1]. Capsular mesenchymal stem cells continuously enter the layer and zG cell lineage[1–3], while fully differentiated zG cells exit the zG layer, transdifferentiating into bona-fide corticosterone-producing zona fasciculata cells[4]. Notably, the zG layer anatomically organizes its cell collectives into multicellular rosette structures, a higher order cyto-architecture observed during organogenesis[5–7] and a key structural unit that facilitates postnatal adrenal glomerular morphogenesis. Interestingly, in contrast to rosettes facilitating embryonic tissue development that are rapidly deconstructed[5], rosettes in the adrenal zG persist throughout adulthood[5] and therefore likely enable ongoing tissue remodeling and/or rapid, diet-driven zG layer contraction and expansion throughout life[8,9]. Whether the rosette structure itself plays a role beyond tissue remodeling and homeostasis remains unknown.

The persistence of the zG layer rosette into adulthood potentially provides the structural basis for cell-cell communication among rosette members. Yet, several observations have supported the opposing conventional view that zG cells are quiescent and autonomous[10,11]. First, numerous electrophysiological studies of dispersed zG cells isolated from human, bovine and rodent glands (i.e., cells removed from the rosette) demonstrate that isolated zG cells maintain a steady hyperpolarized membrane voltage that is merely baseline shifted by hormones (i.e., Angiotensin II, Ang II) or drugs (e.g., potassium channel blockers)[12–18]. Consistent with this quiescent behavior, Ang II evokes in most zG cells a large, transient rise in intracellular calcium that is followed by a small, sustained increase[17]; brief and slow calcium oscillations are elicited only in a small percentage of cells[17,19–21]. Second, zG cells appear to lack the common substrates for cell-cell communication. For example, zG cells lack gap junctions that support electrical communication among cells[10,11].

Whereas these lines of evidence suggest quiet cellular autonomy, observations of zG cells retained within their native rosettes reveal a different innate cellular behavior. We previously demonstrated by patch-clamp electrophysiology that cells within the mouse zG layer produce robust and prolonged voltage oscillations[22] that can drive rhythmic calcium currents. These data confirmed and extended previous findings obtained by sharp electrode recordings of feline adrenal slices[23] and are in agreement with oscillatory calcium activity previously observed in slices[24]. Collectively, these findings raise the intriguing possibility that the rosette structure may foster zG cell communication by entraining synchronized oscillatory activity.

Motivated by these key findings and the premise that multicellular assemblies often do not simply reflect the additive behavior of single cells, we studied the ensemble signaling behavior of individual zG cells within the community structure of the rosette. We measured intracellular calcium, the critical signal that drives the production of aldosterone, in adrenal tissue slices expressing the genetically encoded calcium reporter, GCaMP3. We find that Ang II, the major regulator of aldosterone production from the zG layer, elicits stereotypic oscillatory zG cell calcium bursts whose occurrence is concentration dependent. However, this dependence does not manifest as a simple change in oscillation frequency as is primarily observed with autonomous electrical oscillators. Instead, we show that while zG oscillation frequency is fixed across Ang II concentrations, oscillatory events are organized into sustained bursts of activity, the occurrence of which changes with concentration. The number of these bursts correlates strongly with the production of aldosterone. Finally, we provide evidence for the hypothesis that the adrenal rosette, a dynamic multi-cellular arrangement of zG cells, functions as a network oscillator in which constituent cells communicate to produce bursts of activity that are coordinated.

## Results

### Ang II generates periodic Ca$^{2+}$ Signals in zG Cells.
We used transgenic approaches to study the regulation of intracellular calcium of zG cells within their native rosette structure. Mice in which Cre recombinase was targeted to the Cyp11b2 (AS, aldosterone synthase) genomic locus[4] were crossed with floxed GCaMP3 mice to generate a mouse line (AS$^{Cre/+}$::ROSA26-$^{floxGCaMP3/HZE}$, zG-GCaMP3) in which GCaMP3 was targeted to zG cells. We used wide-field imaging to capture calcium-dependent fluorescent signals in the zG-layer of adrenal slices acutely prepared from zG-GCaMP3 mice. An exemplar slice (Fig. 1a) shows how the dark nuclei and the bright cytosolic fluorescence enhance the visualization of individual cells within the rosette structure of the zG layer. Because Ang II is the major regulator of zG cell activity, we recorded the calcium activity of individual zG cells before and during stimulation with Ang II.

We observed few calcium signals among zG cells during basal, unstimulated conditions. However, 3 nM Ang II evoked robust, periodic calcium oscillations in all recorded zG cells (Fig. 1b, c; see Supplementary Video 1). Calcium oscillations began within 90 s and reached a steady oscillatory state within 5 min of Ang II exposure, with a prominent ~0.5 Hz frequency (Fig. 1b), as reported by Fourier transformation of the raw signal. To complement this analysis, the peak of each oscillatory calcium signal (referred to here as calcium spike) within a slice was detected (see "Methods") and is presented in raster plot format (Fig. 1c, left panel). Applying this approach to all recorded zG cells demonstrated unambiguously in this exemplar slice that 3 nM Ang II evoked a highly stereotyped response; all cells oscillated within a narrow frequency range (<2 Hz, Fig. 1c). In addition, the distribution of inter-calcium spike intervals from all analyzed zG cells was normally distributed with a mean value of 2.2 s (Fig. 1d), consistent with the dominant 0.5 Hz frequency revealed by Fourier analysis.

Next, we characterized the dose-dependency of Ang II-evoked activity of zG cells. The activity of four representative cells from each hormone concentration shows that overall activity increased with dose (Fig. 2a). To quantify activity, we determined the mean calcium spike rate (i.e., number of spikes per minute) of each analyzed zG cell within a slice and then averaged rates across slices from multiple animals (Fig. 2b). Both a dose- and time-dependence were observed; zG cell calcium spikes were initiated with a shorter latency and were more numerous with increasing concentrations of Ang II (Fig. 2a, b). Consistent with electrophysiological recordings of zG cells[22], Ang II-evoked calcium oscillations were dependent on Ca$_v$3 calcium channels, as such activity was abolished by the pan Ca$_v$3 family antagonist, TTA-P2[25] (Fig. 2c, d). By contrast, blocking L-type calcium channels (Cav2.1) with nifedipine[26] had no effect on Ang II-evoked calcium oscillations (Fig. 2e). Curiously, blocking large-conductance, Ca$^{2+}$-activated K$^+$ channels with iberiotoxin[27] resulted in a more rapid onset of evoked oscillations (Fig. 2f), presumably reflecting a decreased oscillatory threshold caused by a basal, depolarizing shift in zG cell membrane potential. In comparison, blocking small-conductance, Ca$^{2+}$-activated K$^+$ channels with apamin[27], produced a marked-prolongation of many but not all oscillatory events (Supplementary Fig. 2a), consistent with blocking a hyperpolarizing activity early in the repolarization phase of the Ca$^{2+}$ oscillation. Preincubation with cyclopiazonic acid (CPA), an inhibitor of SERCA ATPase[28], did not alter Ang II-evoked zG calcium oscillations, indicating that the generation of calcium events was not dependent on endoplasmic reticulum calcium stores (Fig. 2g) or the small expected change in membrane voltage generated by the opening of Orai channels (single channel conductance 10–25 fS)[29]. Together, these data

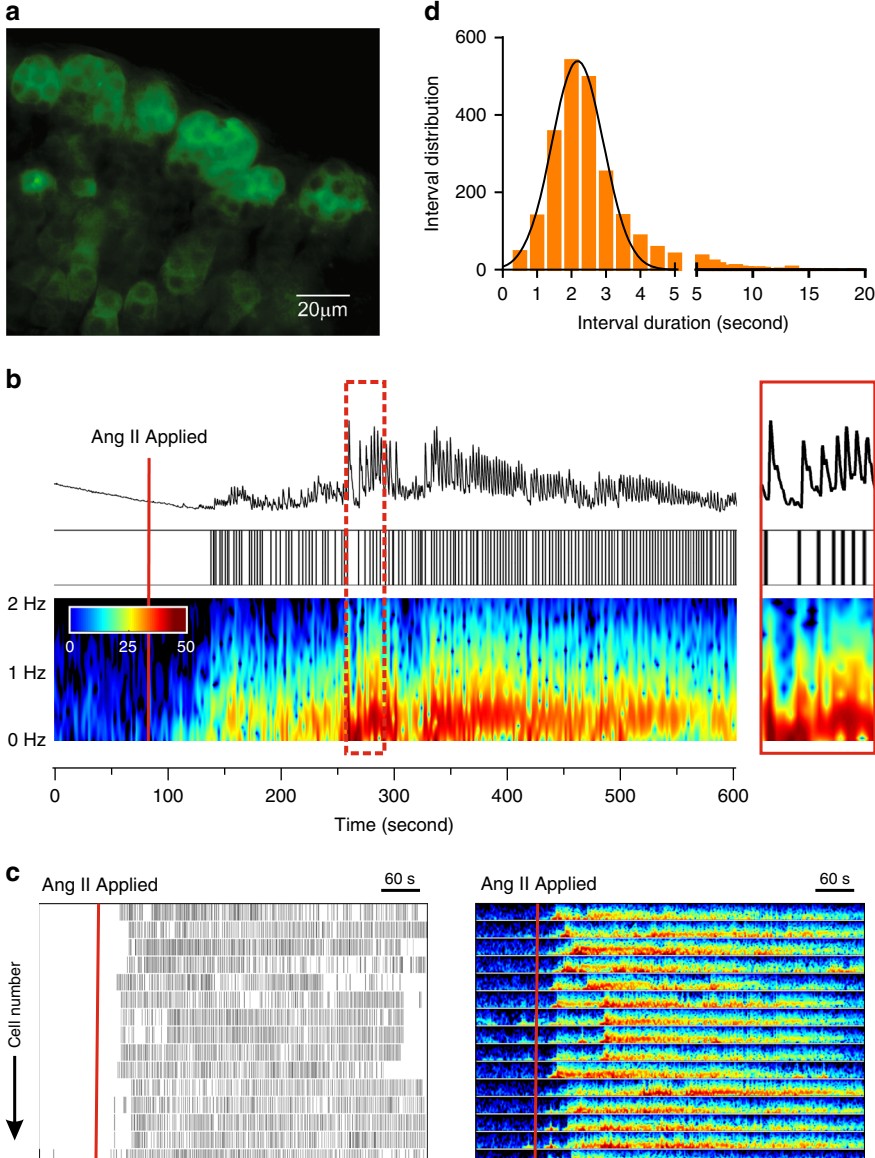

**Fig. 1 Ang II-elicited calcium signaling in zG cells expressing GCaMP3. a** Photomicrograph (63 × magnification) of adrenal slice from a zG-GCaMP3 expressing mouse before Ang II stimulation. Individual cells within rosettes can be distinguished by their dark nucleus, surrounded by cytosolic GCaMP3. Scale bar: 20 μm. **b** Representative trace of fluorescence intensity (top) imaged in a zG cell before and after stimulation with 3 nM Ang II (red line, 90 s after start of record) and corresponding raster plot (middle) of calcium spikes captured at 20 Hz for 10 min. (bottom) Spectrogram of oscillation frequency after Fourier transformation. A magnification of the region indicated by dashed red rectangle is provided on the far right. **c** Raster plot of calcium spikes (left) and corresponding spectrogram of oscillation frequencies (right) from all cells measured in a single adrenal slice. Solid red line indicates when 3 nM Ang II was added to the bath solution. Scale bar: 60 s. **d** Frequency distribution of inter-calcium spike intervals (0.1 min bins) elicited by 3 nM Ang II is normally distributed (mean ± SEM [Hz]: 0.519 ± 0.006, $n = 24,052$). Source data are provided as a Source Data file.

suggest that in the rosette, Ang II evokes graded oscillatory calcium responses that resemble previously described voltage oscillations[22].

Upon closer inspection of calcium spike raster plots, we observed that lower Ang II concentrations produced robust, albeit infrequent, oscillatory bursts in a small subset of zG cells (Fig. 2a). To quantify the number of active zG cells (three or more calcium spikes) across concentrations, we recorded the activity of each slice at a single, submaximal dose of Ang II (50 pM, 300 pM, or 3 nM) and then in response to a maximal dose of 1 μM. By normalizing the number of active cells observed at submaximal doses to the number of active cells at 1 μM, we determined that the percent of activated zG cells within a slice increases with Ang II dose (Fig. 2h). 50 pM Ang II evoked calcium oscillations in

~20% of zG cells, whereas 3 nM Ang II elicited oscillations in nearly all Ang II-reactive zG cells. Combined, these data indicated that within the zG layer Ang II dose-dependently increases both the number of active cells and their activity.

**Burst structure of zG cell calcium oscillations**. As the macroscopic structure of active zG cells appeared similar across Ang II doses (i.e., prolonged, cohesive oscillatory bursts), we next established objective criteria to identify bursts (see Fig. 2a). We first binned calcium inter-spike intervals of zG cells recorded at each dose of Ang II from multiple experiments (e.g., 300 pM: Fig. 3a) and plotted an inter-spike interval histogram. Similar to 3 nM Ang II application, the resultant histogram across all 300 pM Ang II-treated slices revealed a prominent peak at 2 s (cf.

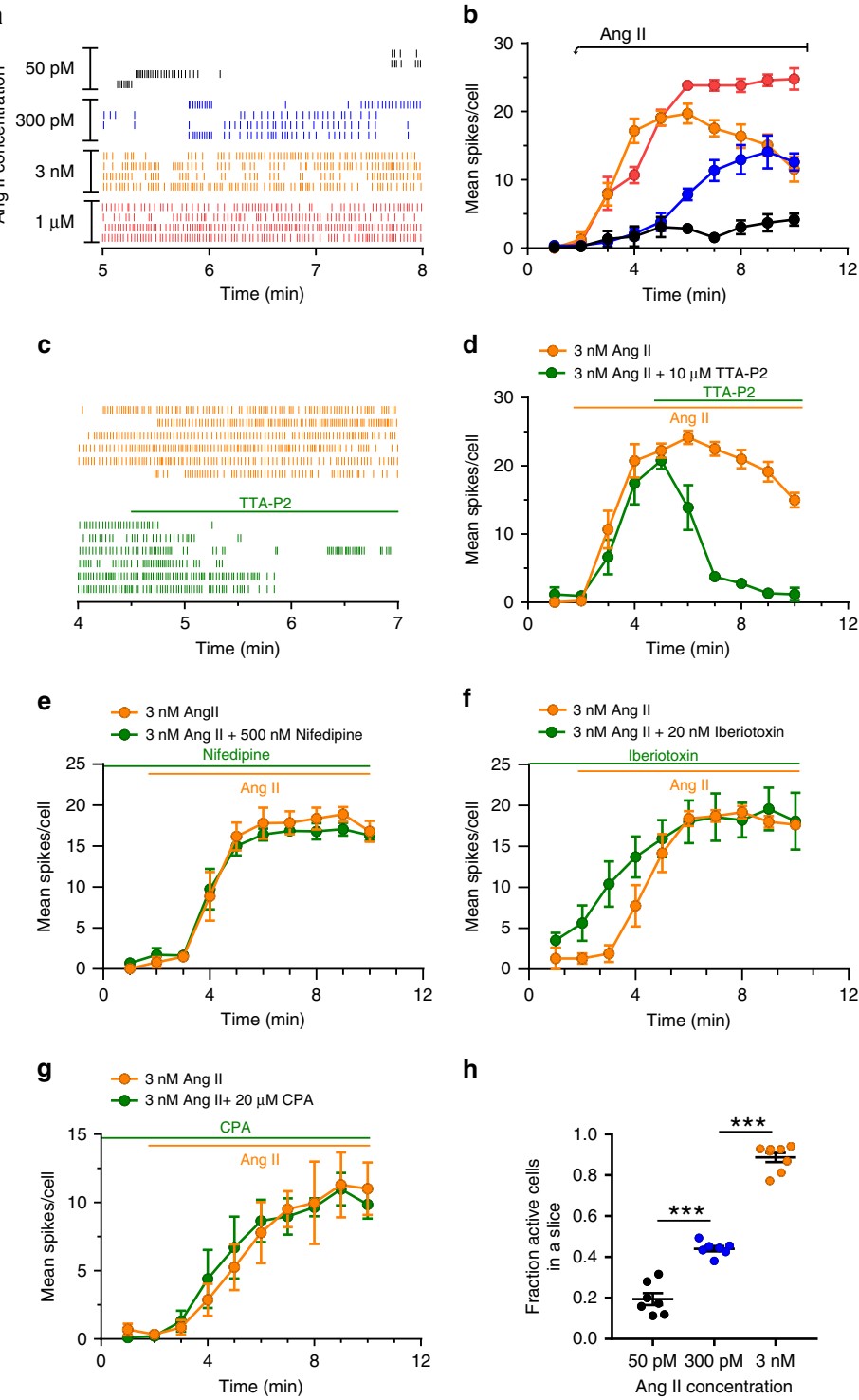

Figs. 1d, 3a) and a smaller, secondary population of spike intervals (>5 s) that reflected the duration of inactivity between two successive bursts. To capture the decay associated with the longer inter-burst intervals, the distribution was fitted with the sum of a Gaussian and an exponential function. This model fitting assumes that inter-spike intervals within a burst are normally distributed, whereas burst occurrence follows a Poisson process with an exponential inter-burst interval distribution. The intersection of these functions determined the threshold value (Fig. 3a, inset) that was used to separate (a) intra-burst calcium spike intervals

within a single, cohesive burst of activity, from (b) inter-burst calcium spike intervals associated with two successive bursts (Fig. 3b). Strikingly, application of this thresholding strategy to zG cell activity on a per-concentration basis showed similar distributions of intra-burst calcium spike intervals (Fig. 3c, bottom). Calculation of the mean spike intra-burst interval in a slice revealed these values to be invariant with Ang II dose (Fig. 3c, top). To evaluate the robustness of this unanticipated result, an alternative thresholding approach was used[30] (Supplementary Fig. 1). In this model, the threshold is defined as the sum of two fitted

**Fig. 2 Calcium spikes elicited by Ang II are dose dependent and require Ca$_V$3 channel activity. a** Representative raster plot of calcium spikes after treatment with 50 pM, 300 pM, 3 nM, or 1 μM Ang II. **b** Mean spikes/cell dose dependently increases with Ang II (50 pM: $n = 5$ slices, 94 cells; 300 pM: $n = 6$ slices, 247 cells; 3 nM: $n = 8$ slices, 165 cells; 1 μM: $n = 4$ slices, 90 cells); 2-way ANOVA: $P < 0.0001$ indicating an overall effect of time and dose. **c** Raster of representative cells and **d** mean spikes/cell with application of the Ca$_V$3 channel antagonist TTA-P2 (10 μM). TTA-P2 (applied at 4.5 min) inhibits calcium spikes elicited by 3 nM Ang II (applied at 1.5 min). TTA-P2: $n = 4$, 96 cells; No TTA-P2: $n = 4$, 90 cells; 2-way ANOVA: $P < 0.0001$ for overall effect of time and drug. **e** 500 nM nifedipine ($n = 4$ per group) had no significant effect on 3 nM Ang II induced activity (2-way ANOVA: $P = 0.942$). **f** However, 20 nM Iberiotoxin resulted in a more rapid onset of spikes, although the overall effect of drug ($P = 0.270$) or drug × time ($P = 0.071$) was not found to be significant across the 10 min period ($n = 4$ per group, 2-way ANOVA). **g** 20 μM CPA did not inhibit calcium spikes, indicating intracellular calcium stores are not required for calcium oscillations. CPA: $n = 5$ slices, 89 cells; No CPA: $n = 5$ slices, 106 cells; 2-way ANOVA: $P = 0.94$ for CPA effect. **h** Fraction of cells that responded to submaximal Ang II dose compared to active cells at 1 μM Ang II; mean ± SEM: 50 pM: 0.19 ± 0.03, $n = 7$ slices; 300 pM: 0.44 ± 0.01, $n = 7$ slices, 3 nM: 0.89 ± 0.02, $n = 8$. One-way ANOVA, $P < 0.001$; Tukey's multiple comparisons test: 50 pM vs 300 pM: ***$P < 0.0001$, 50 pM vs 3 nM: ***$P < 0.0001$, 300 pM vs 3 nM: ***$P < 0.0001$. **b**, **d**–**h** Mean data from each mouse were represented as a single point in calculating N/experimental condition. Lines and symbols are color coded according to experimental condition; 50 pM: black, 300 pM Ang II: blue, 3 nM Ang II: orange, 1 μM Ang II: red, 3 nM Ang II + drug (10 μM TTA-P2, 500 nM nifedipine, 20 nM iberiotoxin, or 20μM CPA): green. Source data are provided as a Source Data file.

Gaussian distribution means (see "Methods"). Application of this algorithm yielded similar threshold values and comparable conclusions about burst structure.

Although spike intervals within a burst were similar across four orders of Ang II concentration, the number of evoked bursts per zG cell was concentration-dependent. Thus, 50 pM Ang II evoked 1–3 bursts (25–75% values) in active zG cells during an 8.5-min period, whereas 1 μM Ang II evoked 9–15 bursts (25–75% values) during a comparable period (Fig. 3d). A clear dose-dependency emerged (Fig. 3e) when averaging the number of bursts per cell. Accordingly, inter-burst intervals were reciprocally shortened with Ang II concentration (Supplementary Fig. 3a, b) Subsequent hormone secretion experiments using a radioimmunoassay (RIA) for hormone detection demonstrated that aldosterone output from the adrenal slice is highly correlated with zG cell burst number (Fig. 3f). Thus, in aggregate, our data indicate that the primary dose-dependent action of Ang II is to increase burst occurrence, not spike frequency, and that burst occurrence drives Ang II-stimulated aldosterone secretion.

Next, we evaluated the properties of individual zG cell bursts across the various Ang II concentrations. Several key features of zG cell bursts were also dose invariant; neither the duration of individual bursts (Fig. 4a, b), nor the number of calcium spikes per burst (Fig. 4c, d), varied with dose. However, consistent with brief inter-burst intervals observed at high agonist concentrations (Supplementary Fig. 3a, b), the latency to periodic calcium activity varied with dose. Latency was shortest with 3 nM and 1 μM Ang II, relative to 50 pM and 300 pM Ang II (Fig. 4e). Notably, the epochs between successive zG cell bursts were not completely quiescent during exposure to lower Ang II concentrations. Instead, these inter-burst periods often were punctuated with intermittent, aperiodic calcium spikes (i.e., singlet calcium spikes) that were most prevalent during 50 pM Ang II application (Fig. 4f, g). Together, these data indicate that bursts evoked by Ang II are very stereotypic with a constant mean duration and a uniform mean number of events.

**Functional clustering of zG cells within adrenal rosettes**. A cursory inspection of Ang II-evoked zG cell activity suggested that the calcium oscillatory behavior among zG cells in close proximity was similar. To objectively evaluate this possibility, we used a functional clustering algorithm[31] (FCA, see Methods) to identify zG cells with synchronous activity patterns. In brief, the FCA pairs cells that produce similar patterns of activity and evaluates the statistical significance of such synchronous pairings by comparing the observed, biological spike trains to numerous, randomly generated surrogate spike trains. Groups of active cells with synchronous activity were assigned to functional clusters

and are represented here as a uniformly colored branch on a dendrogram (Fig. 5a).

Applying the FCA to zG cell calcium spike trains recorded during 3 nM Ang II consistently yielded clusters of cells with relative synchronous activity patterns. Figure 5a shows a representative dendrogram in which zG cell clusters are color-coded, and the corresponding activity rasters for zG cells included in the dendrogram (Fig. 5b). The color-coded activity rasters reveal distinct activity patterns within each functional cluster of zG cells, as defined by the FCA. Next, we localized the anatomical location of each zG cell within the adrenal slice and discovered that functionally paired cells appeared to be situated within the same rosette (Fig. 5c). Notably, across all Ang II doses ~40–80% of active cells belonged to a functional cluster (Fig. 5d).

Unambiguous, visual identification of a complete rosette can be challenging in acute adrenal slices. Therefore, we sought objective criteria to evaluate the anatomical relationships among zG cells within a functional cluster (i.e., within clusters) versus those assigned to different clusters (i.e., between clusters) by the FCA. The centroid coordinates of each region of interest (ROI) used to measure the fluorescence intensity of individual zG cells defined the $X$–$Y$ location of the center of each zG cell within the time-lapsed image. These coordinates were used to determine the Euclidian, pairwise distances of all zG cells within the slice, which were then assigned to one of two groups: (1) within the same functional cluster (i.e., within clusters), or (2) in different functional clusters (i.e., between clusters). zG cell–zG cell distances derived from all 3 nM Ang II experiments are shown in Fig. 5e. The corresponding means from each experiment are shown in Fig. 5f. As is evident, zG cells within functional clusters are in close proximity, relative to zG cells in different clusters. The mean Euclidian, pairwise distance for pairs within clusters was ~10 μm, a value that roughly corresponds to the distance between the centers of two adjacent zG cells situated within a rosette. Pairwise distances within versus between clusters were significantly different at 1 μM and 300 nM Ang II, with a trend at 50 pM (Supplementary Fig. 4). Together, these data reveal a role for the rosette in the coordination of oscillatory calcium signals evoked by Ang II.

**Phase analysis of zG cells within rosettes**. We next applied independent analyses to further support the hypothesis that zG cells within the same rosette produce similar patterns of activity. We systematically evaluated the temporal relationships of calcium spikes produced by all zG cells within a slice, again in a pairwise manner. Specifically, we measured the extent of phase-locking of calcium oscillations across all possible zG cell pair combinations. Two cells which exhibit oscillatory dynamics can be considered phase-locked if their phase difference remains

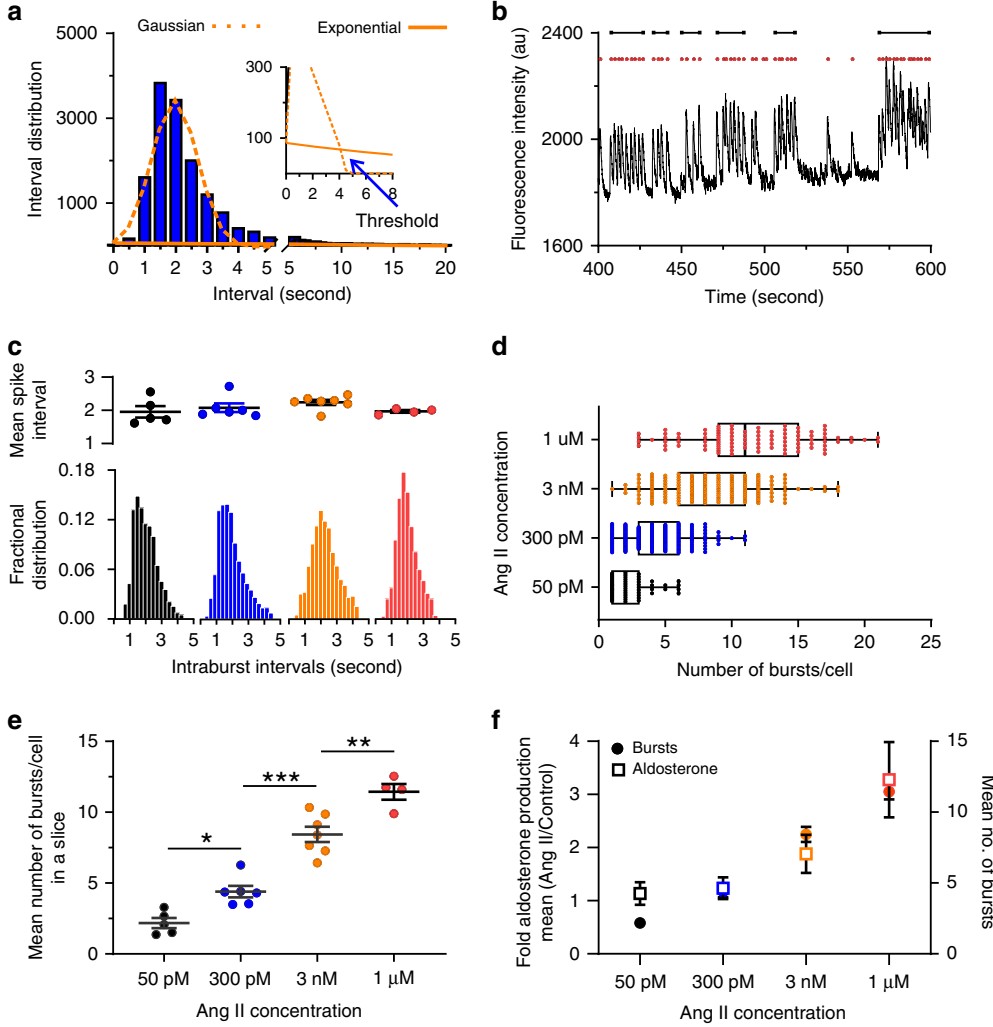

**Fig. 3 Ang II dose-dependently increases the number of bursts per cell. a** Histogram of binned calcium spike intervals across all cells stimulated with 3 nM Ang II and fitted to a Gaussian function (orange dotted line) and exponential function (orange solid line). The intersection of these two functions (blue arrow) represents the threshold for discriminating between intra-burst spikes and inter-burst spikes. **b** Example of trace data from a zG cell stimulated with 300 pM Ang II and the subsequent spike/burst assignment. Calcium spikes (red dots) are grouped within a burst (black lines) if the distance (time) from the previous spike is less than the calculated threshold. To qualify as a burst, the spike train must contain a minimum of three consecutive sub-threshold spikes. **c-f** zG cells stimulated with one of four Ang II doses: 50 pM ($n = 5$, 59 cells), 300 pM ($n = 6$, 217 cells), 3 nM ($n = 7$, 163 cells) or 1 µM ($n = 4$, 95 cells). **c** Top Mean spike interval per cell in a slice; no significant differences among groups were found (Kruskal–Wallis, $P = 0.2071$). **c** Bottom Fractional distribution of intra-burst calcium spike intervals was not dependent on Ang II dose, however (**d**, **e**) number of bursts per cell is concentration dependent (**d**: median value [25–75%]: 50 pM: 2 (1–3); 300 pM: 4 (3–6); 3 nM: 8 (6–11); 1 µM: 11 (9–15) (**e**): means ± SEM [bursts/cell]: 50 pM: 2.17 ± 0.36, 300 pM: 4.41 ± 0.41, 3 nM: 8.44 ± 0.54, 1 µM: 11.44 ± 0.55) 1-way ANOVA $P \leq 0.0001$; Bonferroni's test: 50 pM vs 300 pM (*$P = 0.029$); 300 pM vs 3 nM (***$P < 0.0001$); 3 nM vs 1 µM (***$P < 0.0001$); non-linear regression fit: $R^2 = 0.91$, $EC_{50} = 808.9$ pM. **f** Fold increase in aldosterone secretion in adrenal slice preparations after stimulation with one of four Ang II concentrations (left y-axis, plotted relative to unstimulated secretion, $n = 5$ for each concentration) correlates with the dose-dependence of burst number/cell (right y-axis). Linear regression analysis of bursts vs fold secretion yielded an $r^2$ of 0.88. Mean data from each mouse were represented as a single point in calculating N/experimental condition. **c-f** Bars, symbols are color coded according to experimental condition; 50 pM: black, 300 pM Ang II: blue, 3 nM Ang II: orange, 1 µM Ang II: red. Source data are provided as a Source Data file.

constant over time. This state also requires the mean period of both cells to be equal. Two uncoupled cells cannot achieve a phase-locked state since small natural differences in their oscillatory periods will eventually cause their respective phases to diverge. In our phase analysis, we calculated the timing of each calcium spike produced by one zG cell (i.e., test cell), relative to the periodic spiking of a second, reference zG cell. The location of each test spike was defined according to the normalized spiking period of the reference zG cell (i.e., within a 360° period). If the two zG cells produce calcium spikes simultaneously, then their activities are considered to be phase-locked and characterized by a 0° phase difference (i.e., in phase).

If one zG cell consistently produces a spike precisely midway into the inter-spike period of a second zG cell, then the two zG cells are phase-locked but characterized by a 180° phase difference (i.e., out of phase). Phase locking across the complete spike trains produced by two zG cells can be determined by measuring the standard deviation of all calculated phase differences; coordinated, phase-locked cells are characterized by a low phase standard deviation, as this indicates that the phase-difference between elements are roughly constant over time.

Phase analysis revealed that activity patterns among zG cells within the same rosette are similar, relative to zG cells residing in different rosettes. Phase relationships are readily observed when

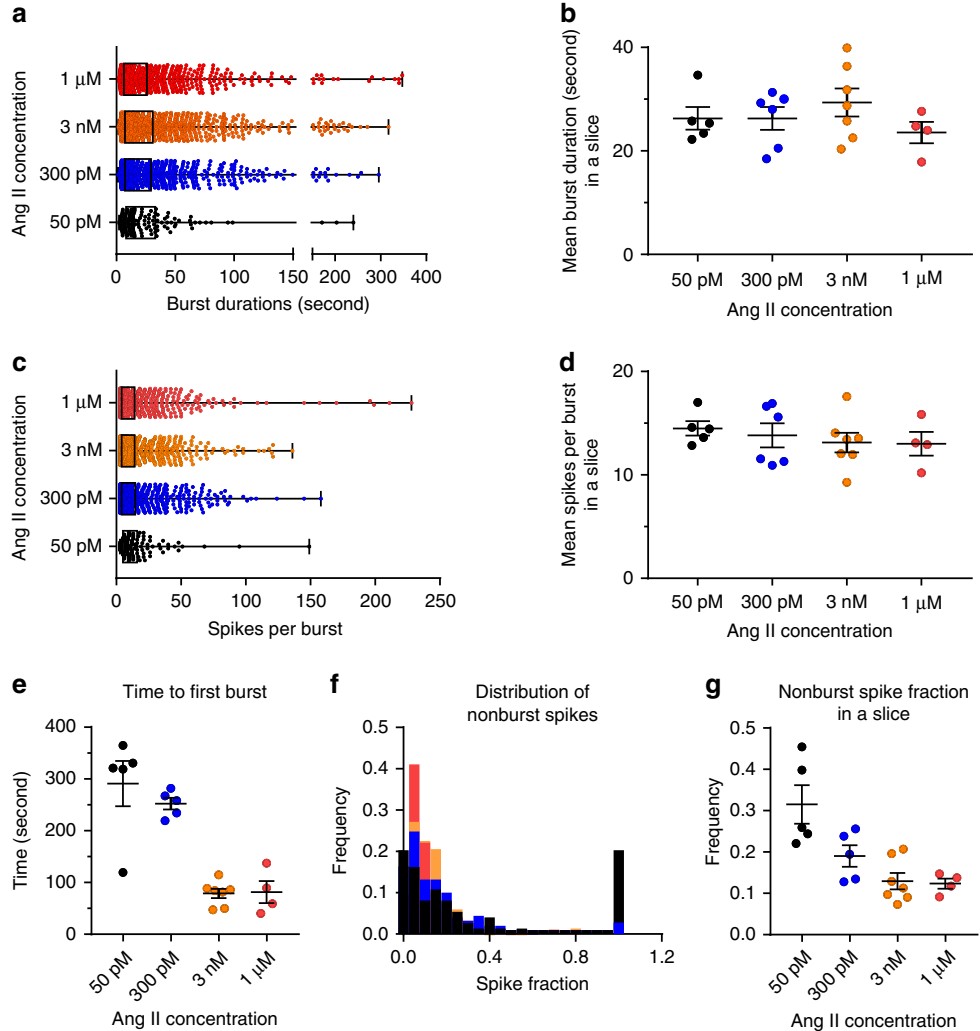

**Fig. 4 Burst analysis of Ang II-elicited calcium spikes.** Burst duration and number of spikes/burst are consistent across Ang II concentrations. **a** Box plot of individual burst duration, median value (25–75%): 50 pM: 14.6 (8.2–14.6); 300 pM: 13.9 (7.3–29.3); 3 nM: 13.9 (7.3–30.8); 1 μM: 11.2 (6.3–26.0); **b** means ± SEM [s]: 50 pM: 26.3 ± 2.2; 300 pM: 26.3 ± 2.2; 3 nM: 29.4 ± 2.7; 1 μM: 23.6 ± 2.1; Kruskal–Wallis test: $P = 0.4783$; **c** box plot of individual burst spikes, median value (25–75%): 50 pM: 6 (4–14), 300 pM: 6 (4–14), 3 nM: 7 (4–14.5), 1 μM: 9 (5–16); **d** means ± SEM; Kruskal–Wallis test: $P = 0.65$). **e** However, both the time it took for a cell to respond to Ang II (time to first burst, means ± SEM (s): 50 pM: 291 ± 44; 300 pM: 252 ± 11; 3 nM: 79 ± 9; 1 μM: 81 ± 21; Kruskal–Wallis test: $P = 0.002$, Dunn's multiple comparison test: 50 pM vs 3 nM: $P = 0.006$; 50 pM vs 1 μM $P = 0.0398$; non-linear curve fit: $R^2 = 0.77$, $IC_{50} = 366.1$ pM) and fraction of nonburst spikes (**f, g**) decreased with Ang II dose (**f** distribution of the fraction of non-burst spikes in individual cells, 50 pM: $n = 74$; 300 pM: $n = 250$; 3 nM: $n = 151$; 1 μM: $n = 95$; **g** means ± SEM: 50 pM: 0.32 ± 0.05; 300 pM: 0.19 ± 0.03; 3 nM: 0.13 ± 0.02; 1 μM: 0.12 ± 0.01; Kruskal–Wallis test, $P = 0.008$, Dunn's multiple comparison test: 50 pM vs 3 nM $P = 0.0078$, non-linear curve fit: $R^2 = 0.77$, $IC_{50} = 366.1$ pM). Unless otherwise noted, for Ang II concentrations: 50 pM: $n = 5$, 135 cells; 300 pM: $n = 6$, 1017 cells; 3 nM: $n = 7$, 1398 cells; 1 μM: $n = 4$, 1098 cells. **b, d, e, g** Mean data from each mouse were represented as a single point in calculating N/experimental condition. Bars and symbols are color coded according to experimental condition; 50 pM: black, 300 pM Ang II: blue, 3 nM Ang II: orange, 1 μM Ang II: red. Source data are provided as a Source Data file.

plotting the phase standard deviations across all zG cell pairs within a slice (Fig. 6a). To visualize phase relationships among cells within and between rosettes, the X- and Y-axes of the resultant matrix plots are ordered intentionally according to rosette membership. Notably, zG cells within the same rosette exhibit more phase-locked behavior (blue/green hues) than those residing in different rosettes (red/yellow hues). White boxes demarcate individual rosettes selected by a researcher blind to the FCA/phase analysis. Calculating the cumulative distribution of the standard deviations of the phase difference and their respective means for all Ang II doses revealed that within rosette phase standard deviations (solid lines) were well separated from between rosette standard deviations (dotted lines) (Fig. 6b, c). Thus, by two independent measures—the FCA and phase analysis —zG cells within a rosette exhibit functional coupling.

**Extrinsic factors regulate zG calcium oscillations.** Thus far, our data revealed that zG cells within the same rosette appear to produce coordinated patterns of oscillatory calcium activity. However, zG cells lack the most common molecular machineries necessary to support cellular communication. The zG layer is devoid of gap junctions as demonstrated by multiple independent assays (freeze fracture, transmission electron microscopy, live cell imaging, dye transfer, pharmacological blockade and mRNA assays[10,11]). In addition, there is scant evidence for direct control of zG cell activity by purinergic signals, in contrast to the medullary zone where purinergic receptors (P2Xx- and P2Yx-subtypes) and secretory vesicles are abundant[32,33]. In the absence of these modalities, we hypothesized that the close proximity of zG cells situated within a rosette could provide an avenue for zG cell communication. Putatively, this could occur via ephaptic transmission when the electrical

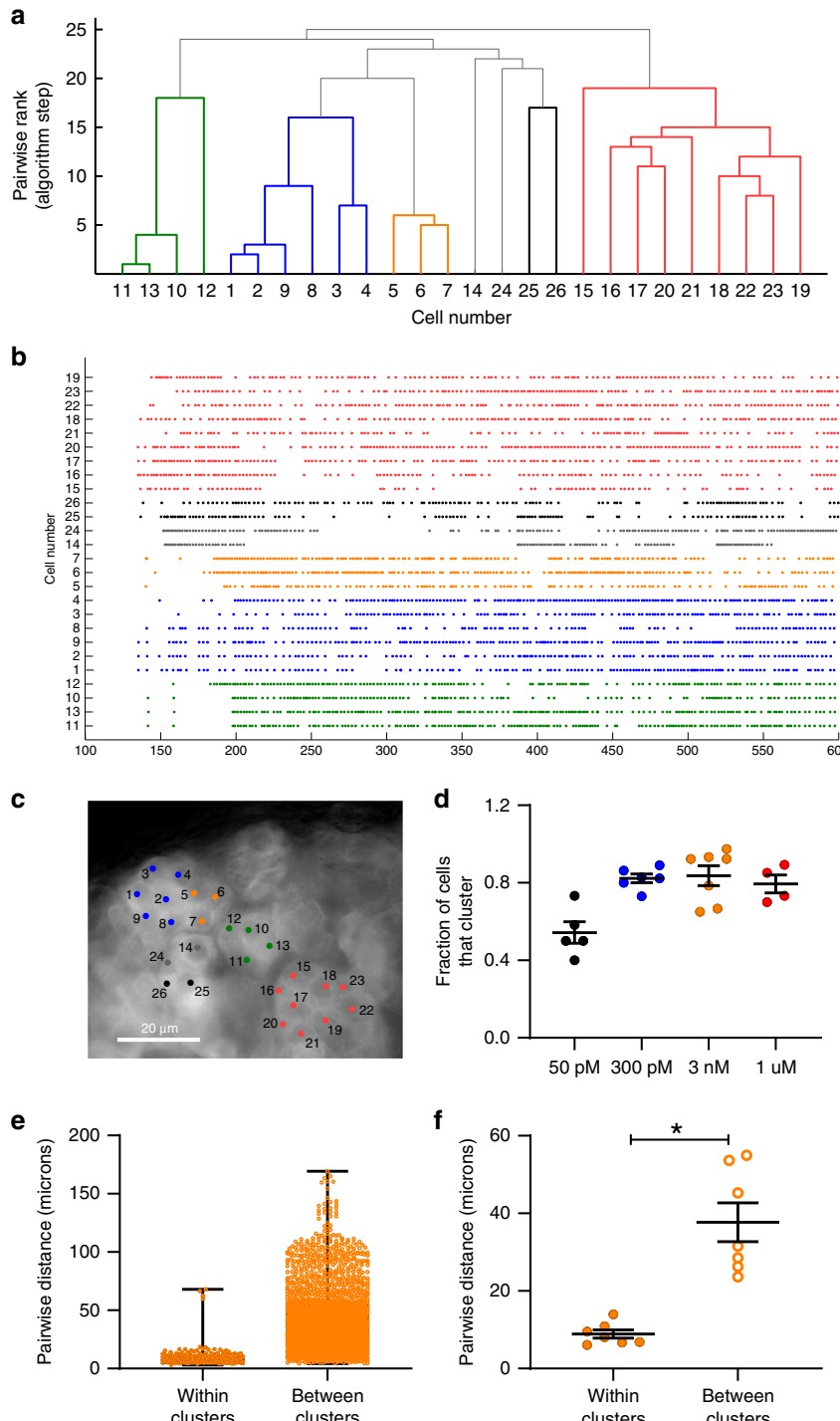

**Fig. 5 zG cells within rosettes functionally cluster with neighboring cells after stimulation with 3 nM Ang II.** Representative data from an adrenal slice stimulated with 3 nM Ang II. **a** Dendrogram output from the functional clustering algorithm (FCA) for ROIs shown in **c**, indicating cell pairings based on calcium spike synchrony. Cells (ROIs) are numbered for identification (x-axis), and are ranked pair-wise (y-axis) by the FCA based on the mean least (temporal) distance between spikes of each cell; those that pair best are ranked lowest on the dendrogram and grouped together (i.e., 1 on the y-axis), while pairs with more diverse temporal spike relationships appear higher on the dendrogram. Each color represents a group of cells that significantly paired with each other (i.e., a cluster). **b** Raster plot of calcium spikes by time (s) across all cells marked, arranged by groupings and the (**c**) corresponding ROIs selected in the slice experiment. **d** Across Ang II concentrations, most cells were assigned to a cluster group; however, a significantly smaller fraction of cells cluster at 50 pM (means ± SEM: 50 pM (black, n = 5 slices): 0.543 ± 0.056; 300 pM (blue, n = 6 slices): 0.823 ± 0.023; 3 nM (orange, n = 7 slices): 0.837 ± 0.051; 1 μM (red, n = 4 slices): 0.795 ± 0.046; 1-way ANOVA: P = 0.0009, Bonferroni: P ≤ 0.017 for 50 pM vs 300 pM, 3 nM, and 1 μM). **e, f** The distance between cells that cluster is markedly smaller than cells that do not pair (**e**, box plot of all paired distances across 7 experiments after 3 nM Ang II, median value (25–75%) [μm]: clustered: 7.2 (5.7–10.8), n = 226; non-clustered: 37.0 (21.8–56.4), n = 1827). **f** Means ± SEM (μm) per adrenal experiment (3 nM Ang II, n = 7): clustered: 8.9 ± 1.1; non-clustered: 37.7 ± 5.0; Mann–Whitney test: *P = 0.0156. Mean data from each mouse were represented as a single point in calculating N/experimental condition. Source data are provided as a Source Data file.

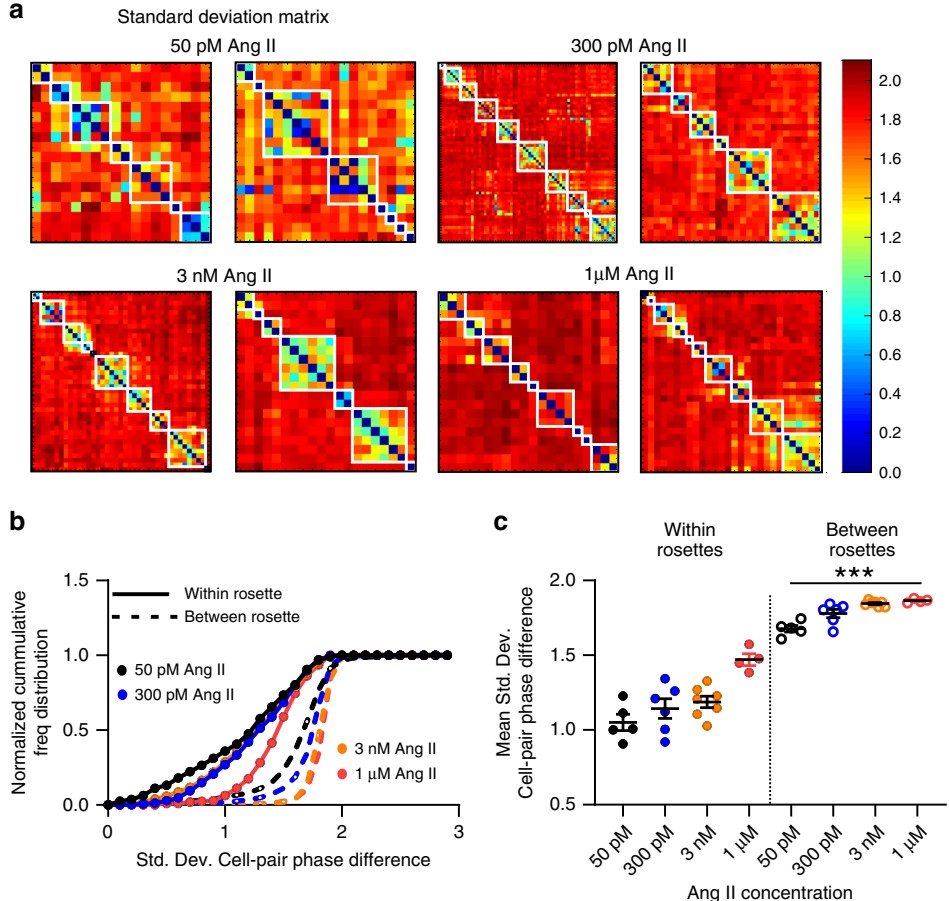

**Fig. 6 Ang II-induced calcium spikes have phase-stability among rosette members. a** Two representative phase analyses of calcium spikes shown in matrix plots for each Ang II concentration. Individual cells are ordered along the *X*- and *Y*-coordinates; each square of the matrix represents the standard deviation of phase difference (PD-STDEV) between two cells and is color-coded (blue: low PD-STDEV; red: high PD-STDEV). White boxes delineate individual rosettes. Dark blue diagonal across each plot corresponds to self-pairing. Phase-locking is greater among cells within rosettes than between rosettes, as indicated by more blue/green hues, a leftward shift in the cumulative phase distribution (**b**) and an overall smaller mean PD-STDEV (**c**) (within-rosette mean PD-STDEV ± SEM: 50 pM: 1.05 ± 0.055, 300 pM: 1.14 ± 0.065; 3 nM: 1.19 ± 0.039; 1 µM: 1.47 ± 0.039; between-rosette mean PD-STDEV ± SEM: 50 pM: 1.68 ± 0.022, 300 pM: 1.78 ± 0.028; 3 nM: 1.85 ± 0.007; 1 µM: 1.86 ± 0.007; 1-way ANOVA $P \leq 0.0001$; within-rosette mean PD-STDEV for each dose differed significantly from every dose between-rosette mean PD-STDEV [***$P \leq 0.037$]). **b**, **c** Lines and symbols are color coded according to experimental condition; 50 pM: black, 300 pM Ang II: blue, 3 nM Ang II: orange, 1 µM Ang II: red. Source data are provided as a Source Data file.

fields of one cell influence closely apposed cells[34], or via mechanical transmission when membrane stress imposed on morphologically tethered cells is shared[35,36]. We therefore tested whether close proximity regulates zG calcium bursts.

Cadherins represent a family of transmembrane proteins mediating junctional connections that tether cells together[37]. N- and K-cadherins (N/KCADs) play well-established roles in cell-cell adhesion and cadherins are the major constituents of the adherens junctional complex (AJ) in many tissues[38,39]. Both NCADs and KCADs are abundant in rosette structures and are required for rosette formation[5]. Using immunohistochemical approaches, we localized both cadherins to zG cell-containing adrenal rosettes. Consistent with cadherin subcellular localization[40], we detected strong staining of NCADs (magenta, Fig. 7a, top) and KCADs (magenta, Fig. 7a, bottom) at the membrane surface of zG cells. Filamentous actin (F-actin), an additional key component of AJs, was also evident at cell-cell contacts (green) and detected among larger, CAD-positive aggregates (white) found at rosette centers where multiple zG cells interface.

To test for a functional role for AJs in zG calcium activity, we initially evaluated the consequences of weakening NCAD-

mediated cellular junctions. Transcellular ecto-domain (EC) interactions of CAD proteins within AJs are calcium-dependent[41]. Following calcium sequestration, transcellular connections mediated by AJs can be disrupted by anti-ecto-domain antibodies[42]. First, we sought to mildly disrupt CAD-dependent adhesions employing an EGTA-based calcium switch assay, to evaluate the contribution of cellular contacts within the pinwheel structure of the rosette to zG calcium oscillations[42]. Specifically, we compared zG cell activity following a brief incubation with either an antibody directed towards the extracellular domain of NCAD (EC-Ab) or the intracellular domain of NCAD (IC-Ab). The latter served as a control for non-specific disruption as the inaccessibility of the IC-antigenic epitope precludes IC-Ab binding. We also compared these experimental conditions to protocol-matched control conditions in which adrenal slices were incubated only with equivalent sequential changes in extracellular calcium.

EC-Ab incubation reduced zG cell spiking activity. The mean number of spikes produced per minute by zG cells trended lower in adrenal slices incubated with EC-Abs, whereas activity with IC-Ab incubation was equivalent to that of control solutions

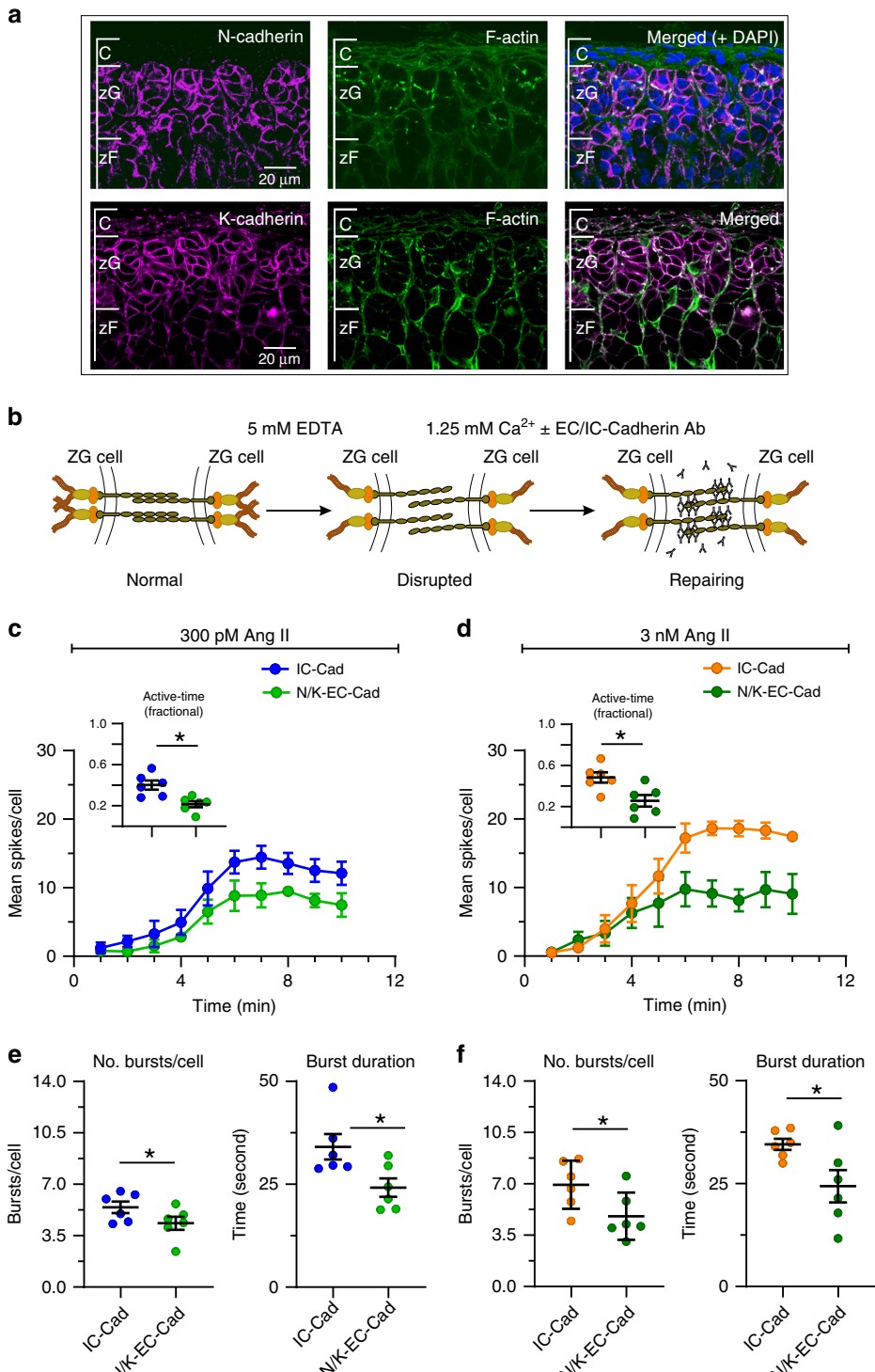

(Supplementary Fig. 5a). This time-dependent trend in cellular activity elicited by EC-antibodies was accompanied by a significant shortening of the burst duration; the average duration was ~30% briefer than burst durations occurring in IC and control incubations (Supplementary Fig. 5b). By contrast, across all conditions the number of bursts produced by zG cells remained unchanged (Supplementary Fig. 5c), as did the intra-burst spike frequency (Supplementary Fig. 5d) and the latency to activity following Ang II application (Supplementary Fig. 5e).

Because N-CADs and K-CADs serve overlapping roles in cellular adhesion[38,43], we speculated that preserved K-CAD

function in the previous set of experiments may have minimized junctional disruption. Therefore, we used a combination of antibodies directed against the extracellular domain of either N-CAD or K-CAD in an EDTA-based calcium switch assay[44] to determine if more aggressive junctional disruption produced substantive changes in zG cell calcium activity (Fig. 7b). We applied this treatment to slices stimulated by either 300 pM or 3 nM Ang II. N/K-CAD disruption reduced the number of calcium events observed at both Ang II concentrations (Fig. 7c, d), leading to a reduction in the fractional active time (insets). The combined N/K-EC antibodies reduced both the number and the duration of

**Fig. 7 Disruption of N-cadherin shortens burst duration without affecting other calcium spike parameters. a** Immunohistochemistry showing N-cadherin (red, upper left), K-cadherin (red, lower left) and F-actin (green, middle) are abundant in the zG layer (right, merged images). Scale bar: 20 μm. **b** Model of calcium switch assay: Disrupting adherins junctional complex by reducing extracellular calcium with EDTA. Anti-N/K-cadherin antibodies (EC-Abs) binds to the now exposed extracellular N- and K-cadherin binding sites, inhibiting reformation of strong cell–cell adhesion after calcium is restored. For controls, we substituted an antibody directed to the conserved intracellular domain (IC-Ab) of cadherins. **c–f** Adrenal slices were stimulated with 300 pM (**c, e**) or 3 nM (**d, f**) Ang II after 1.5 min of baseline recording. EC-Abs: $n = 6$; IC-Ab: $n = 6$. **c, d** Mean spikes per 1 min bin over a 10 min period; EC-Abs treated slices had fewer spikes over time for both 300 pM and 3 nM Ang II (2-way ANOVA, effect of antibody: $P = 0.011$ and 0.018, respectively) as well as mean fractional active time (**c, d** insets, mean ± SEM: 300 pM: IC-Ab 0.40 ± 0.04, EC-Abs 0.22 ± 0.03; 3 nM: IC-Ab 0.48 ± 0.05, EC-Abs 0.26 ± 0.06; Wilcoxon rank test, *$P = 0.016$ and 0.016, respectively). **e, f** Reduced activity was due to a decrease in the number of bursts per cell (**e, f** left panels; mean ± SEM: 300 pM: IC-Ab 5.43 ± 0.39, EC-Abs 4.35 ± 0.44; 3 nM: IC-Ab 6.94 ± 0.67, EC-Abs 4.8 ± 0.66; Wicoxon rank test; 300 pM: *$P = 0.047$, 3 nM: *$P = 0.016$) and a shortening of burst duration (**e, f** right panels; mean ± SEM: 300 pM: IC-Ab 34.1 ± 3.10, EC-Abs 24.2 ± 2.25; 3 nM: IC-Ab 34.54 ± 1.36, EC-Abs 24.38 ± 3.94; Wilcoxon rank test; 300 pM: *$P = 0.016$, 3 nM: *$P = 0.047$). Mean data from each mouse were represented as a single point in calculating N/ experimental condition. **c, d** Lines and symbols are color coded according to experimental condition; IC-Ab + 300 pM Ang II: blue, EC-Abs + 300 pM Ang II: light green, IC-Ab+ 3 nM Ang II: orange, EC-Abs + 3 nM Ang II: dark green. Source data are provided as a Source Data file.

Ang II-evoked bursts (Fig. 7e, f), relative to their IC-antibody controls. In addition, N/K-EC antibodies reduced the number of cells that clustered in response to 300 pM or 3 nM Ang II-stimulation (Fig. 8a). However, those zG cells that did cluster following the calcium switch assay remained in close proximity (Fig. 8b) and produced phase-locked calcium spikes within their respective rosettes (Fig. 8c). Importantly, both baseline and Ang II-stimulated aldosterone production from acute adrenal slices was reduced by treatment with EC-directed antibodies, relative to IC-directed antibodies (Fig. 8d).

## Discussion
Here we demonstrate that zG cells situated within their native rosettes produce robust, sustained calcium oscillations in response to the primary regulator of aldosterone production, Ang II. We show that the intensity of Ang II-evoked calcium oscillations is strongly dose-dependent, and that such dependency does not involve a simple increase in oscillation frequency with increasing Ang II concentration. Instead, we find that the highly stereotyped, macroscopic pattern of zG cell calcium activity—the zG calcium burst—is the fundamental unit of zG cell activity that is more frequently evoked with increasing Ang II dose. This unit of signaling activity correlates significantly with the magnitude of aldosterone production. We also show that cellular junctions between closely-apposed zG cells support zG calcium bursts, indicating that the adrenal rosette provides an anatomical substrate for the functional modification of zG cell activity. Moreover, our functional cluster and phase analyses provide evidence that calcium oscillations produced by zG cells within a rosette are highly coordinated and form a connection-based network of oscillating cells. Collectively, these results motivate us to rethink the molecular and cellular processes that govern the zG cell calcium signal required for aldosterone production.

Many cells are endowed with the capacity to produce highly periodic ionic (and, therefore, voltage) fluctuations on their own[45–48]. To achieve such rhythmogenesis, these autonomously oscillating cells must express the appropriate combination of voltage- and time-dependent conductances to sustain cellular oscillations without input from other cells. As the underlying conductances are voltage-dependent, autonomous oscillations are observed only within a range of membrane potentials. For example, the hyperpolarized-activated cation current ($I_h$) and the low threshold, T-type calcium current ($I_t$), two voltage-dependent conductances, are sufficient to endow thalamocortical neurons with the capacity to produce 1–2 Hz voltage oscillations[45,49]. However, such oscillations are abolished when these neurons are sufficiently hyperpolarized or depolarized to membrane potentials that cause sustained $I_h$ and/or $I_t$ channel closure[45,50]. Moreover, within the rhythmogenic voltage range, oscillation frequency

generally increases with depolarization[45,46,50–52], a result of shifting the voltage-dependent activation and inactivation gating kinetics[53] of the underlying conductances.

Several observations indicate that zG cells of the adrenal gland are not strict autonomous oscillators. First and foremost, experimental preparations of isolated zG cells do not produce robust, sustained voltage oscillations, even in the presence of Ang II[18,54]. While the agonist depolarizes isolated zG cells[13] and can evoke brief oscillatory bursts of activity in a small subset of isolated cells[55,56], isolated zG cells removed from their native rosette do not oscillate. In stark contrast to this behavior is the finding that robust, highly stereotyped zG cell voltage and calcium oscillations are reliably observed in situ[22,24].

A second piece of evidence indicating that zG cells are not strict autonomous oscillators is our finding that calcium spike frequency is dose-invariant; irrespective of Ang II concentration, the frequency of zG cell calcium spikes within a burst was consistently ~0.5 Hz. As Ang II depolarizes zG cells and the strength of this depolarization is dose-dependent[18,54], the frequency of calcium oscillations produced by an autonomously oscillating zG cell would also be dose-dependent. However, no dependency was observed. Therefore, either the sensitivity of our assay is insufficient to resolve voltage-dependent frequency changes in an autonomously oscillating zG cell, or the zG cell is not an autonomous oscillator; in aggregate, the evidence supports the latter hypothesis.

If zG cell oscillatory activity is defined by the presence or absence of factors normally found within the adrenal tissue, then one primary goal is the identification of such factors. Full realization of this goal is confounded by multiple lines of evidence demonstrating that zG cells do not contain the standard molecular components that support extrinsically-driven voltage oscillations. zG cells neither express gap junctions that can mediate electrical transmission between cells[10,11], nor do they contain the morphological substrates associated with chemical transmission (i.e., vesicles)[32,33]. What zG layer feature then provides a platform for Ang II-evoked zG cell oscillations and their coordination?

Here, we provide evidence that the rosette, with its abundant expression of cadherins, provides the scaffold necessary for the expression of robust voltage and calcium oscillations evoked by Ang II. Cadherins regulate cell-cell contact[57] and their trans-bond engagement produces adhesive tension that recruits the catenins (p120-catenin, β-catenin, α-catenin) to the cadherin intracellular domain[58]. Subsequent mechanosensitive recruitment of additional actin binding proteins anchors the cadherins to the actin cytoskeleton[59]. Thus, within a rosette cadherins serve to interconnect the cytoskeletons of constituent cellular members[60]. By regulating interfacial tension[6,60], cadherins can also indirectly modulate the activity of tension-sensitive ion channels and G-protein coupled receptors[61] raising the intriguing possibility that

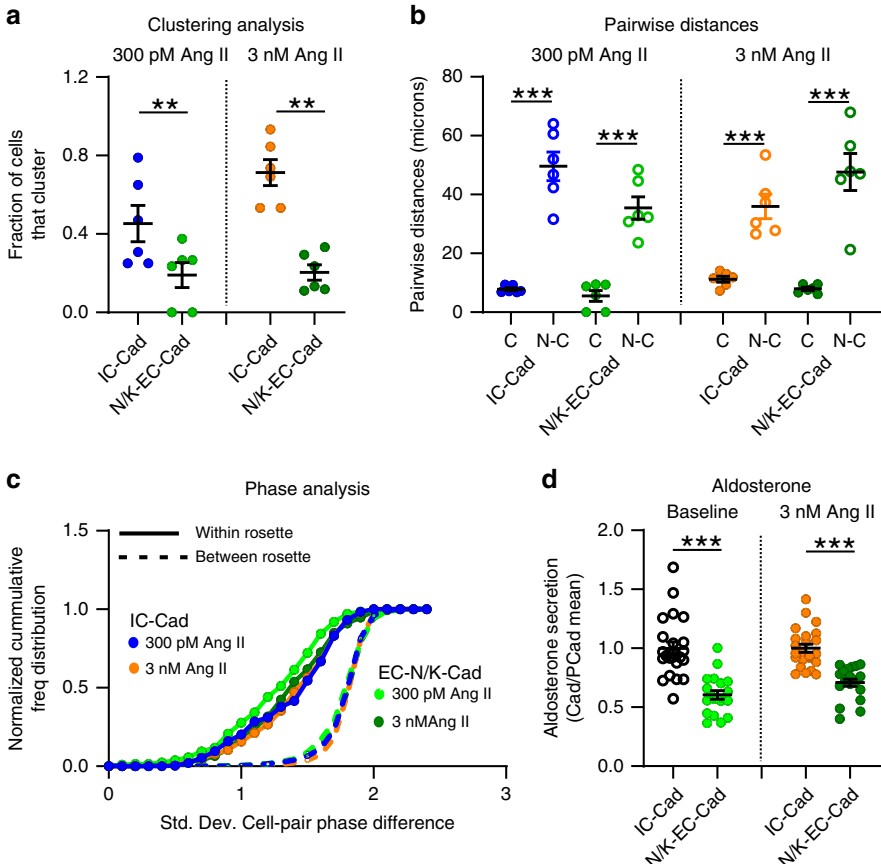

**Fig. 8 Cadherin disruption reduces both the number of clustered cells and the production of aldosterone. a** Significantly fewer cells clustered after N/K EC-Abs treatment when stimulated with 300 pM or 3 nM Ang II (means ± SEM: 300 pM Ang II: IC-Ab 0.45 ± 0.19, EC-Abs 0.09 ± 0.06; 3 nM Ang II: IC-Ab 0.71 ± 0.20, EC-Abs 0.07 ± 0.04; 1-way ANOVA, 300 pM Ang II: *$P = 0.026$; 3 nM Ang II: ***$P < 0.0001$). **b** However, cells that remained in functional clusters were proximate (C: clustered; N-C: non-clustered; means ± SEM: 300 pM Ang II: IC-Ab-C 7.77 ± 0.50, IC-Ab-N-C 49.60 ± 4.89, EC-Abs-C 5.55 ± 1.85, EC-Abs-N-C 35.44 ± 3.78; 3 nM Ang II: IC-Ab-C 11.20 ± 1.02, IC-Ab-N-C 35.96 ± 4.13, EC-Abs-C 7.95 ± 0.58, EC-Abs-N-C 47.65 ± 6.30; 1-way ANOVA, 300pM Ang II: IC-Ab C vs C-N ***$P < 0.0001$, EC-Abs C vs C-N ***$P < 0.0001$; 3 nM Ang II: IC-Ab C vs C-N ***$P = 0.0005$, EC-Abs C vs C-N ***$P < 0.0001$) and **c** produced phase-locked calcium oscillations (smaller PD-STDEV). **d** PD-STDEV Notably, aldosterone production was reduced after N-/K-cadherin disruption, both at baseline and following 3 nM Ang II stimulation (means ± SEM: baseline: IC-Ab 1.00 ± 0.05 [$n = 23$], EC-Abs 0.60 ± 0.04 [$n = 20$]; 3 nM Ang II: IC-Ab 1.00 ± 0.04 [$n = 23$], EC-Abs 0.70 ± 0.03 [$n = 20$]; 1-way ANOVA, baseline: ***$P < 0.0001$; 3 nM Ang II: ***$P < 0.0001$). Each data point represents aldosterone secreted from slices within a well standardized to the mean production in control (IC-Ab) wells. Lines and symbols are color coded according to experimental condition; IC-Ab + 300 pM Ang II: blue, EC-Abs + 300 pM Ang II: light green, IC-Ab + 3 nM Ang II: orange, EC-Abs + 3 nM Ang II: dark green. Source data are provided as a Source Data file.

"intrinsic" or autonomous oscillatory activity observed in previous patch-clamp recordings[22] may have been provoked by mechanical stress from the recording pipette.

Cadherins junctions are highly dynamic structures that can reorganize after disruption[40,62]. Calcium binding rigidifies the cadherin extracellular domains that promote cellular adhesion[63]. Conversely, lowering calcium disrupts binding and adhesion of all cadherins[40]. NCAD is the least calcium-sensitive member among the type I and type II CAD subfamilies [NCD: $K_D \sim 0.65$ mM[41]], and implies that NCADs can rapidly respond to physiological reductions in extracellular calcium. For example, reduced calcium concentrations are predicted to occur in synaptic clefts during high frequency neuronal firing[64], a stimulus that triggers extensive synapse remodeling and plasticity, changes that are prevented by neuronal NCAD disruption using specific EC-domain antibodies[62,65]. Using specific EC-domain antibodies, we demonstrate here that NCAD disruption alone produced only modest changes in calcium bursting implying that cell adhesion within the adrenal rosette may not be regulated dynamically by extracellular calcium under physiological conditions. However, because combined N/K-CAD antibody treatment significantly

reduced activity, we conclude that adhesion stability within the rosette is nonetheless important for zG calcium signaling and endocrine function. Whether the remaining antibody-insensitive oscillatory activity is adhesion-dependent awaits further anatomical evaluation of the extent of antibody-induced junctional disruption. In sum, we show that CADs localized within the adrenal rosette functionally support burst initiation (burst number) and termination (burst duration), in contrast to Ang II whose singular function is to regulate burst initiation.

Surprisingly, elevated Ang II doses reliably evoked coordinated oscillatory activity among adrenal rosette members. If primarily sensitive to calcium spike frequency, then the FCA cluster analysis likely would have assigned all cells recorded within a slice to one large functional group, as calcium spike frequency was similar across rosettes. However, FCA-defined functional clusters primarily included zG cells with similar burst on- and off-set times. Intriguingly, these clusters also mapped onto individual rosettes as confirmed by phase analysis, suggesting that the rosette serves to coordinate sustained and concurrent bursting activity among resident zG cells. Thus, rosette-intrinsic factors (CADs) appear to ensure that extrinsically-initiated activity (Ang II) begins at the

same time and is coordinated. In the absence of a single cell assay for aldosterone, we speculate that within a presumed heterogeneous population of cells (at varied stages of maturation and zG to zF lineage conversion)[66] cellular communication within the structure of the rosette promotes the recruitment and entrainment of less responsive cells to guarantee a coordinated tissue output. Our quantitative analysis of Ang II-evoked mean oscillatory activity supports this speculation as mean burst number per cell and the level of aldosterone output were highly coordinated across Ang II doses. In conclusion, our analysis provides a framework for understanding rosette-based mechanisms driving calcium activity within the zG layer, and thus may broadly inform treatments for aldosterone-associated hypertension, a more prevalent disease than originally acknowledged[67,68].

## Methods

**Mice.** AS (aldosterone synthase)$^{+/Cre}$ mice, where Cre expression is under the control of the aldosterone synthase promoter[4], were bred to B6;129S-*Gt(ROSA) 26Sor*$^{tm38(CAG-GCaMP3)Hze}$/J (purchased from Jackson Laboratory, stock 014538) generating mice with zG specific GCaMP3 expression (zG-GCaMP3 mice). Mice were group housed in a temperature and humidity controlled room on a 12:12 light:dark cycle. For aldosterone secretion experiments, C57Bl6 mice were purchased from The Jackson Laboratory.

**Adrenal slice preparation.** Male zG-GCaMP3 mice between 40 and 100 days of age were anesthetized with ketamine (15 mg, i.p.), and adrenals dissected. Adrenals were sectioned (60–70 μm) in ice-cold PIPES incubation buffer (in mM: 20 PIPES, 117 NaCl, 3 KCl, 1 CaCl$_2$, 1 MgCl, 25 D-Glucose, 5 NaHCO$_3$, pH 7.3). Sections were kept at 37 °C for 30 min, and then allowed to return to room temperature for the duration of the experiment. Some sections were pre-treated for ~20 min with either 20 μM CPA, 500 nM nifedipine, 10 nM iberiotoxin, 100 nM apamin, or vehicle in incubation buffer, and drugs were included in perfusion buffer throughout acquisition. Sections used for cadherin disruption experiments were maintained at 37 °C and incubated in buffer containing 5 mM EDTA for 45 min, followed by a 2-h incubation with anti-N-cadherin (Sigma Aldrich, C3865) and anti-K-cadherin antibodies (Fisher Scientific, MA1-06305) targeted to extracellular domains (EC-Abs), or an anti-Pan-cadherin antibody that target the highly-conserved intracellular domain (IC-Ab, Sigma Aldrich, C1821)[44]. This final incubation buffer contained no EDTA, 1.25 mM calcium, and incubation antibody concentrations of 1:25 (for each EC-Abs) or 2:25 (IC-Ab).

**Aldosterone secretion.** Adrenal glands from male C57Bl6/J mice (45–80 days old) were sectioned (60 μm thick) as described under adrenal slice preparation. In a 24-well plate, adrenal sections were divided 2–4 per well in 500 μl secretion media: MEM with Hank's balanced salts (Sigma, M4780) supplemented with (in mmol/L): 1 L-Glutamine, 1 CaCl$_2$, 30 HEPES, 2 MgCl$_2$, 12.8 D-Glucose, 5.24 NaHCO$_3$, 12-mg/L bovine insulin, 0.12% ascorbic acid, 0.1% horse serum, and 0.1% BSA at 37 °C. For cadherin disruption experiments, sections were incubated for an additional 45 min in secretion media containing 5 mM EDTA, followed by 2 h in secretion media with antibody, as described above. Wells were sampled (150 μl) after 15 min of incubation (baseline), and the sample volume replaced with fresh media containing Ang II or vehicle. After 30 min, a final sample was collected. Aldosterone content of sampled media was measured by radioimmunoassay (RIA, Tecan US, Inc., MG13051).

**Immunostaining of adrenal slices.** Adrenal glands from female C57Bl6/J mice (8–10-week old) were dissected, trimmed of surrounding fat tissue, rinsed in PBS, cut in halves with a surgical blade, and fixed in 4% PFA at 4 °C for 1 h. PFA-fixed adrenals were washed in PBS and embedded in 4% low-melting-temperature Sea-Plaque Agarose (Lonza, 50100), and sectioned on a vibratome at a thickness of 100 μm. Slices were washed with 0.1% Tween-20 in PBS for 10 min for three times, and blocked in 5% Normal Goat Serum, 1% Bovine Serum Albumin, 0.1% Tween-20 in PBS for 1 h at RT with gentle rocking. Slices were incubated with a polyclonal rabbit anti N-cadherin (Novus Biologicals, NBP2-38856) or K-cadherin (Abcam, ab133632) primary antibody diluted 1:100 in blocking solution at 4 °C overnight. After three 20 min washes with 0.1% Tween-20 in PBS, adrenal slices were incubated with an Alexa Fluor 488-conjugated goat anti rabbit IgG secondary antibody (Invitrogen, A-11034) diluted 1:100 in 0.1% Tween-20 in PBS for 2 h at RT. For F-actin staining, Alexa Fluor 647-conjugated Phalloidin (Invitrogen, A22287) was added to the secondary antibody mix for the last 30 min at final concentration of 1:100. For nuclear staining, DAPI was added to secondary antibody mix for the last 5 min at final concentration of 1:500–1:1000. Slices were then washed with 0.1% Tween-20 in PBS for 20 min for three times and mounted on Superfrost Plus slides (Fisher Scientific, 12-550-15) with Prolong Gold (Life Technologies, P36930). Images were acquired using a Zeiss LSM710 confocal microscope with a ×63/1.4 oil immersion Plan-Apochromat objective and adjusted for brightness and contrast using ImageJ.

**Image acquisition.** Adrenal slices were imaged in a PIPES buffer (in mM: 20 PIPES, 122 NaCl, 3 KCl, 1 CaCl$_2$, 1 MgCl, 25 D-Glucose, 0.1% BSA, pH 7.3) on a Zeiss Axio-Examiner microscope fitted with a xenon arc light source (Lamda DG4, Sutter Inst.) with a custom-made stage configured for a perfusion system, and images acquired at 20 Hz with an sCMOS camera (Hamamatsu Orca-Flash 4.0) using HCImageLive software (Hamamatsu) or Slidebook 6 (Intelligent Imaging Innovations). Each experiment started with a 90 s baseline acquisition period before adding hormone/drug through the perfusion system. Experiments lasted between 10 and 15 min and were saved as high quality multipage tiffs for analysis. The following drugs were used: Angiotensin II (50 pM–1 μM, Fisher Scientific, TTA-P2 (10 μM, Merck), cyclopia-zonic acid (CPA, 20 μM, Cayman Chemicals), nifedipine (500 nM, Sigma), apamin (100 nM, Tocris) and iberiotoxin (10 nM, Cayman Chemicals).

## Data analysis

**Signal extraction.** Fluorescence intensity is extracted from ROIs (regions of interest) using Caltracer software (Yuste lab, Columbia U., Matlab) to produce a time series of average ROI intensity. ROI placement is performed manually on identified cells in each slice using a stacked image (multipage tiff) from an early segment of the optical time-series, and registered to correct for movement artifacts over time when necessary. Image output levels were adjusted to include all information-containing pixels, and balance and contrast adjusted to reflect true rendering as much as possible for ROI placement (e.g., Fig. 1a). Additional ROIs are placed as background controls.

**Event detection.** Calcium transient events in each time-series are identified using custom Matlab software combining an automated event detection algorithm with manual corrections for erroneous or missing events.

**Cell activity.** Calcium activity is presented in raster plots and quantified to determine mean events/cell, mean frequency/cell, and bursting activity. Periodicity is determined by Fourier transform and by inter-event interval. To assess manipulation-dependent changes in cellular activity, Calcium events are binned into time segments (e.g.,1-min bins) and averaged across cells to establish slice behavior; each slice represents a sample size of one prepared from a single animal.

**Burst detection.** Events were grouped into bursts as follows: The inter-event interval distribution was fitted with a mixture of Gaussian and exponential distributions (see equation for Supplementary Fig. 1a). Our method was based on the assumption that (i) bursts occur infrequently but independently of each other at a constant rate and can be modeled as a stationary Poisson process (which yields exponentially distributed inter-arrival times) and (ii) events within bursts have periods drawn from a Gaussian distribution. The inter-event interval value at the intersection of the Gaussian and exponential component of the distribution was taken as the maximum inter-event interval within a burst.

A separate method for fitting the inter-event interval distribution with a series of Gaussian distributions based on Bodova et al.[30] was also implemented (see equation for Supplementary Fig. 1b). Bodova et al.'s method was based on the assumption that there is a subthreshold oscillation of two states (quiescent versus spiking) that transitions with fixed probability (as a Markov chain) and have periods drawn from two different Gaussian distributions. In this model, the sum of the two fitted Gaussian distribution means were used as the maximum inter-event interval within a burst.

Events were then grouped as bursts if there were at least three consecutive events with inter-event intervals all greater than the threshold described above. The resulting burst detection were then plotted and visually inspected for verification of the algorithm.

**Functional clustering algorithm**. Within each slice experiment, a functional clustering algorithm was used to analyze event patterns among cells completely blind to a cells location[31]. In brief, cells are sequentially paired according to the degree of similarity between event patterns. The relationship of each cell-pair is assessed by calculating the minimum duration between an event in one cell and the nearest event in a second cell. The mean minimum duration of all events in the pair defines activity similarity and is used to assign functional groups. Statistical significance of detected clusters is determined by comparisons to surrogate data sets, and pairing considered significant if $P < 0.001$. Finally, degree of similarity and cluster membership are depicted in a dendrogram. These analyses are computationally intensive and, therefore, run on UVA's Rivanna High Performance cluster.

**Phase distribution**. Phase distribution analysis assesses time-locked behavior between cell pairs by measuring the timing variability (i.e., standard deviation) of spikes generated by two cells[69]. Phase relationships with a low standard deviation indicate that two cells are phase-locked (i.e., fixed temporal relationship). Non-time-locked signals yield a random distribution of phases with high standard deviation. Color-coded matrix plots reveal all relationships.

**Statistics**. Data were presented as mean ± standard error or median followed by 25–75 percentile values. When comparing means, cells from a single slice were considered a single data point when calculating $N$ for statistical analysis, and only one slice was used per animal for a given experimental condition. Differences in means were considered significant if $P < 0.05$ using a two-tailed test unless otherwise noted. We used parametric ANOVAs (1- or 2-way) with Bonferroni (1-way) or Tukey's multiple comparison test (2-way) to compare means if the data passed the Shapiro–Wilk normality test, otherwise we used non-parametric Kruskal–Wallis test with Dunn's multiple comparison test. To determine the burst modification underlying activity reduction in adherin junction disruption experiments, we used a one-tailed Wilcoxon matched-pairs signed rank test (IC-Ab vs EC-Abs per mouse adrenal, $n = 6$, $P < 0.05$). Prism 7 software (Graphpad) was used for all statistical analysis.

**Study approval**. These studies have complied with all relevant ethical regulations for animal testing and research. All experiments were performed in accordance with the National Institutes of Health Guide for the Care and Use of Laboratory Animals and approved by the University of Virginia Animal Care and Use Committee.

**Reporting summary**. Further information on research design is available in the Nature Research Reporting Summary linked to this article.

## Data availability

All datasets generated during and/or analyzed during the current study are available from the corresponding author on reasonable request. The source data underlying Figs. 1d, 2b, d–h, 3c–f, 4a–g, 5d-f, 6b–c, 7c–f, 8a–d, and Supplementary Figs 2b–d, 3a–b, 4, 5a–e, 6 are provided as a Source Data file.

## Code availability

Custom code to generate specific analysis are available from the corresponding author Mark P. Beenhakker, and can be downloaded from https://github.com/cagancayco/ImageAnalysisHub.

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

## Acknowledgements

We wish to acknowledge thoughtful discussions with: S. Feldt Muldoon, E. Hall, and J. Huband during the early stages of this work. This work was supported by grants from the National Institutes of Health awarded to: P.Q.B. (HL-36977, HL-138241), M.P.B. (NS-099586), D.T.B. (DK-100653A), and P.M.K. (T32-GM008328).

## Author contributions

Experiments were conceived and designed by NAG, PMK, PQB, MPB, and DTB. DTB provided $AS^{Cre/+}$ mice. Calcium imaging experiments were conducted by NAG, PMK, and RRM, immunohistochemistry by SL. NAG, CAG, AL, PMK, CC, CGR, MPB, PQB contributed to the analysis of data and interpretation of results. MPB, PQB, NAG wrote the manuscript, with editing contributions from all authors.

## Competing interests

The authors declare no competing interests.
