## [Peer Review File · Nature Communications]

Reviewers' comments:

Reviewer #1 (Remarks to the Author):

Guagliardo and colleagues report on periodic Cav3-dependent calcium spikes and bursts in an angiotensin II (AngII) dependent manner and provide indication that the anatomical structures of “rosettes” can modulate this mode of action. Specifically, the authors demonstrate, that AngII increases the number of bursts, while spike intervals within a burst were similar across applied Ang II concentration. Utilizing antibodies directed towards the extracellular domain of N-cadherins (NCAD) that decreased AngII dependent spike activity (i.e. shortening of burst duration) they provide evidence for a modulation of this effect through intercellular signaling further fostered by the findings of unguided clustering of cells within close proximity that showed similar pattern of cellular activity. The strength of the study includes the usage of state-of-the-art imaging and electrophysiological methods that provide new insights in to the physiology of adrenal zona glomerulosa cells.

- One of the main limitation includes the lack of information whether the observed structural and functional findings that are restricted on the evaluation of signaling in fact are related to the endocrine function of ZG cells, i.e. aldosterone output. It would be reassuring to see data that provide a correlation e.g. between burst numbers and aldosterone production. While the direct detection of aldosterone on a cellular level is yet out of reach, I wonder, whether the quantification of Cyp11b2 expression levels (by in situ hybridization or immunohistochemistry) could be a way to overcome this weakness

- Data summarized in Figure 5 are not very convincing in comparison to the fold differences and significances that the authors were able to detect in the other experimental settings. In line 205-207 it is wrongly stated that “the mean number of spikes produced per minute by zG cells consistently was lowest in adrenal slices incubated with EC-abs” as Figure legend 5C states that no significant differences had been found between experimental conditions. While I appreciate the tendency of mean events/cell upon incubation with the antibody targeting the extra- and intracellular domain of N-cadherin, the only remaining difference is that on burst duration. Obviously, there are a number of potential explanations, why the difference is minor – including the lack of penetration of the antibody into the tissue section, duration of incubation, binding capacity of the antibody and lack of blockage of other molecules involved in the cell-cell interaction of ZG cells. These potential limitations of the experimental setting should be discussed. Furthermore, it would add strength to the relevance of the “calcium switch assay” if other endpoints could be explored. Specifically, I would be interested, whether N-cadherin dependent disruption of cellular interaction would have effects on functional clustering –as nicely demonstrated in Figure 6.

- a burst has been defined for the examples of 300pM AngII. It would be important to show whether the same applies for other concentration of AngII

- Line 122: not sure, why it is referred to Figure 2B.

Reviewer #2 (Remarks to the Author):

The authors examine functional relationships between calcium activity of adrenal rosette cells. They find that cells display bursting behavior with a frequency that increases with the dose of Ang II. Further, the pattern of the calcium bursts can be clustered into functional groups with similar bursting patterns that are composed of spatially localized cells. I find the manuscript to be well written, the analysis appropriate, the results novel and interesting, and I have only the following minor comments and suggestions for improvement.

When performing the spatial analysis of functional clusters, the authors measure the average pairwise distance between cells within the same cluster and compare it to the pairwise distance between cells in different clusters. While this clearly shows that the functional clusters are spatially localized, it does not necessarily show that they are within the same rosette. For example, one functional cluster could span two rosettes that are located next to each other. I understand that it is difficult to visually identify the rosettes, but it would greatly strengthen the findings if the authors could find more evidence to convince the reader that the functional clusters showed high spatial overlap with rosettes. Potentially the authors could ask an expert blind to the functional findings to partition the image into rosettes, and an overlap measure could be calculated? Even if the partitioning wasn't exact, a high overlap would help to show that it is likely that the functional clusters do correspond to the spatial rosettes. The authors could also calculate the spatial extent (area defined by the convex hull) of the functional clusters and show that this corresponds with the expected spatial extent of a rosette, although this would not prove overlap.

Although this is just terminology, I found it very confusing that the phrases "clustered pairs" and "non-clustered pairs" were used to describe the calculation of distances between cells. The functional clustering algorithm can lead to cells that are assigned to their own cluster (ie, individual cells that are not grouped with other cells – gray cells in Fig 6) which is what the phrase "non-clustered cells" brings to mind. Perhaps the phrases "within cluster pairs" and "between cluster pairs" or something similar can be used to avoid this confusion. Although the authors describe the meaning in the text, simply reading the figure without the corresponding text can be misleading.

I have a similar comment regarding "centroid distance". It is easy to think that this relates to the centroid of the functional cluster, not of individual cells. I think something like "pairwise distance" would be less confusing.

It would be useful to add a few words in line 138 to very briefly describe what the alternative thresholding method is so that the reader doesn't have to reference the SI just to have an idea of what the alternative was.

Reviewer #3 (Remarks to the Author):

Guagliardo et al. studied Ca²⁺ signalling underlying angiotensin II (Ang II) -stimulated aldosterone production from rosettes of adrenal zona glomerulosa (zG) cells. Whereas isolated zG cells are known to respond to Ang II with little Ca²⁺ signalling, presently studied zG cells within rosettes expressing the Ca²⁺ indicator GCaMP3 showed highly reproducible and stereotypic Ca²⁺ responses to a wide range of Ang II concentration. This signalling was characterized by bursts of short Ca²⁺ spikes with a typical inter-spike period of about 2 sec. The bursts were separated by >5 sec periods with no or few irregularly occurring spikes. This Ca²⁺ activity depended on voltage-regulated Ca²⁺ channels and was prevented by a T-type channel blocker but was insensitive to disruption of Ca²⁺ release from the ER by the SERCA inhibitor cyclopiazonic acid. Unlike Ca²⁺ signalling in some other types of cells the inter-spike period was unaffected by the agonist concentration and the number of spikes per burst as well as burst duration were also concentration-independent. However, the lag-time between Ang II addition and the initial Ca²⁺ response decreased with increasing concentrations together with shortening of the inter-burst period resulting in increasing number of bursts with Ang II concentration.

Functional cluster analysis of Ca²⁺ spike synchrony between cell pairs revealed that closely located zG cells that likely belong to the same rosette formation show synchronized Ca²⁺ responses to Ang II that differ from the synchronized responses in other rosettes. Since zG cells lack synchronizing gap junctions and are not known to release factors that might mediate cell synchronization the authors focussed on other possible mechanisms. As N-cadherins (NCADs) are abundant and required for rosette formation, their possible importance for Ca²⁺ synchronization was investigated. A protocol with Ca²⁺ reduction to disrupt NCAD connections was followed by exposure to antibodies against the extracellular domain of NCAD to prevent reconnection upon subsequent restoration of the Ca²⁺ concentration. This approach resulted in somewhat reduced number of Ca²⁺ spikes in response to Ang II, whereas there was no effect when an antibody against the inaccessible intracellular NCAD domain was used. Interestingly, detailed analysis revealed that this effect was solely due to reduction of burst duration, with no effect on intraburst Ca²⁺ spike frequency or lag-time to the first burst. This effect contrasts to the shortening of lag time that was observed with increasing concentrations of Ang II.

The authors conclude that rosette-intrinsic NCAD coupling somehow ensures that Ang II can initiate coordinated Ca²⁺ spiking in zG rosettes to stimulate aldosterone production.

This interesting study is well written and I like the meticulous analysis of the signalling patterns. However, why reproducible Ca²⁺ spike patterns are generated only in zG cells within rosettes, occur in bursts and become synchronized among the rosette cells remains enigmatic.

Major issues

1. Since not only T-type but also L-type Ca²⁺ channels have been implicated in the regulation of aldosterone production the authors should investigate how blockers of these channels affect Ang II-induced Ca²⁺ signalling in zG rosette cells.
2. It is interesting that Ang II concentration does not affect spike frequency but increases the number of bursts. No attempt is made to characterize the underlying ionic events in the bursting phenomenon.
3. Although I like the analysis of signalling patterns I lack information of how well the Ca²⁺ spikes are synchronized between cells. The raster plots do not provide sufficiently detailed information to the reader, and it would be valuable with some discussion about the nature of the synchronization mechanism based on the relationship between individual spikes in different zG cells within a rosette. Does the coupling mechanism provide a basis for the simultaneous occurrence of spikes at random in different rosette cells or are the individual spikes somehow coupled between cells (occur with a certain time difference and/or order)?
4. I am not happy with the statement in lines 267-268 that “cellular junctions between closely-apposed zG cells regulate zG calcium bursts”. In my view the word “regulate” is inappropriate since no regulatory role is demonstrated and the junctions may merely provide sufficient cellular compactness for another synchronizing mechanism.
5. In the same context I find the hypo-osmotic challenge data presented in Supplemental Fig. 3 interesting but difficult to interpret. Since the study deals with Ang II-induced Ca²⁺ responses it is confusing to discuss activation of the AT1-receptor in the absence of ligand.
6. The effect of the calcium switch assay to disrupt cellular junctions is rather modest. Is it possible to somehow quantify the effectiveness of the cell junction disruption efficiency and relate it to the effect on Ca²⁺ signalling?

Minor issues

7. Add time scale to Fig. 1B

8. Clarity in Fig. 2E would be improved by green symbols indicating “3 nM Ang II + 20 μ M CPA” rather than “+20 μ M CPA”.
9. Line 257, with reference to Fig S4 I suppose 300nM is a misprint and should be 300pM.
10. Panel C in Fig 5 indicates 300 nM Ang II but the figure legend states 300 pM.
11. The X-axis in Fig. 6B is unlabelled.

Reviewer 1:

“Guagliardo and colleagues report on periodic Cav3-dependent calcium spikes and bursts in an angiotensin II (AngII) dependent manner and provide indication that the anatomical structures of “rosettes” can modulate this mode of action. Specifically, the authors demonstrate, that AngII increases the number of bursts, while spike intervals within a burst were similar across applied Ang II concentration. Utilizing antibodies directed towards the extracellular domain of N-cadherins (NCAD) that decreased AngII dependent spike activity (i.e. shortening of burst duration) they provide evidence for a modulation of this effect through intercellular signaling further fostered by the findings of unguided clustering of cells within close proximity that showed similar pattern of cellular activity. The strength of the study includes the usage of state-of-the-art imaging and electrophysiological methods that provide new insights in to the physiology of adrenal zona glomerulosa cells.”

Thank you for recognizing the strength of our manuscript.

- 1. “One of the main limitation includes the lack of information whether the observed structural and functional findings that are restricted on the evaluation of signaling in fact are related to the endocrine function of ZG cells, i.e. aldosterone output. It would be reassuring to see data that provide a correlation e.g. between burst numbers and aldosterone production. While the direct detection of aldosterone on a cellular level is yet out of reach, I wonder, whether the quantification of Cyp11b2 expression levels (by in situ hybridization or immunohistochemistry) could be a way to overcome this weakness”.***

We agree that the first version of our manuscript did not include an evaluation of endocrine function (i.e. aldosterone output). We now provide new data to address this deficiency. Using a radioimmunoassay (RIA), we have measured aldosterone output from adrenal slices on a time scale that is similar to our imaging studies. That is, our imaging experiments typically lasted 15 minutes, while our aldosterone secretion experiments lasted 30 minutes. The sensitivity of the RIA, coupled with the need to study aldosterone production from several slices in aggregate, dictated the discrepancy in timing. Nonetheless, we now show a strong correlation between Ang II-evoked burst number and Ang II-induced aldosterone production ($r^2=0.88$, **Figure 3F**).

We have also used our RIA-based endocrine analysis on the anti-cadherin antibody disruption studies (**Figure 8D**). In slices treated with a combination of anti-N- and anti-K-cadherin extracellular domain (EC) antibodies, we show a reduction in baseline and Ang II-stimulated aldosterone production compared to slices treated with an anti-intracellular domain (IC) antibody that recognizes a region common to all cadherins. The decision to include an anti-K-cadherin antibody in our disruption protocol was based on immunological data from our collaborator showing localization of N- and K-cadherins to zG cell rosettes (we provide an image of K-cadherin protein expression (**Figure 7A**) and also refer to a companion manuscript under review at Nature Communications).

- 2. “Data summarized in Figure 5 are not very convincing in comparison to the fold differences and significances that the authors were able to detect in the other experimental settings. In line 205-207 it is wrongly stated that “the mean number of spikes produced per minute by zG cells***

consistently was lowest in adrenal slices incubated with EC-abs” as Figure legend 5C states that no significant differences had been found between experimental conditions. While I appreciate the tendency of mean events/cell upon incubation with the antibody targeting the extra- and intracellular domain of N-cadherin, the only remaining difference is that on burst duration. Obviously, there are a number of potential explanations, why the difference is minor – including the lack of penetration of the antibody into the tissue section, duration of incubation, binding capacity of the antibody and lack of blockage of other molecules involved in the cell-cell interaction of ZG cells. These potential limitations of the experimental setting should be discussed.”

We agree. A significant change in calcium activity (burst duration) was evident with an anti-N-cadherin EC domain antibody alone, but the effect was admittedly modest. As indicated above in Response #1, we amended our cadherin disruption protocol to include both an anti-K-Cad and an anti-N-cad antibody because both K- and N-cadherins are abundant in the zG layer, and the disruption of one alone may be insufficient. This approach yielded a more robust change in calcium activity. Thus, we moved the original, single antibody data set from Figure 5 to supplemental **Figure 5A-E** and now present our new data using the combination antibody strategy in **Figure 7 & 8**. We tested antibody disruption on calcium activity evoked by two concentrations of hormone, 300 pM and 3 nM Ang II, as requested. We summarize our new findings as follows:

- A. At each hormone concentration, N/K-Cad antibody disruption reduced the mean number of events/cell resulting *in a significant* decrease in mean fractional active-time. **Figure 7C, D**
 - B. At each hormone concentration, the fewer events observed during N/K-antibody treatment resulted in a *significant* decrease in burst number as well as a *significant* shortening of burst duration. **Figure 7E, F.**
 - C. We have added a more extensive discussion of the strengths and limitations of these studies in the discussion, page18, lines 370-377.
3. *“Furthermore, it would add strength to the relevance of the “calcium switch assay” if other endpoints could be explored. Specifically, I would be interested, whether N-cadherin dependent disruption of cellular interaction would have effects on functional clustering –as nicely demonstrated in Figure 6.”*

We concur. We summarize our new analyses as follows:

- A. **Functional clustering.** We provide a functional clustering analysis of our new N/K-cadherin disruption data in **Figure 8**. We show in **Figure 8A** that N/K-cadherin disruption *significantly* decreases the number of active cells that cluster (exhibiting relative temporal synchrony), relative to IC-antibody treatment. Cells that remain clustered following antibody treatment remain proximally located to each other **Figure 8B**.
- B. **Phase analysis.** We provide an additional analysis to assess activity coordination (i.e. phase analysis). Unlike the functional clustering algorithm that measures the degree to which cells maintain synchronous activity patterns, phase analysis evaluates the degree to which the temporal relationship of activity patterns generated by two cells is stable. Thus, we define *synchrony* as the simultaneous occurrence of calcium oscillations/events. In contrast, we define *phase-locked* as a fixed temporal relationship between calcium spikes generated by two cells,

but one that does not necessarily indicate synchrony. For example, two cells that consistently spike 200 msec apart are phase-locked, but not synchronized, while two cells that consistently spike 0 msec apart are both phase-locked and synchronized. Phase-locking (i.e. temporal relationship stability) can be evaluated by quantifying the standard deviation in spike latencies between two cells; phase-locked cells are characterized by a low standard deviation.

We now apply the two independent measures of temporal relationships – functional clustering and phase analysis – to our AngII dose-response experiments, as well as to our antibody experiments. Regarding the former, we find that not only do functional clusters map to cells proximately located but that the activity patterns generated by cells within anatomically-defined rosettes are phase-locked; activity patterns among cells in different rosettes are not phase-locked (**Figure 6**). Thus, by two independent measures, zG cell activity within a rosette is coordinated.

Regarding our antibody experiments, we notably find that N/K-cadherin disruption reduces activity synchrony but does not reduce activity phase-locking among cells within a rosette (**Figure 8B, C**). Together these data indicate that cadherin-mediated cell-cell interactions regulate *oscillatory synchrony and the activity level of zG cells*, the latter of which is consistent with the lack of sustained oscillatory behavior of zG cells in dispersed cell preparations.

C. Aldosterone production. Using our RIA assay, we now show that N/K-cadherin disruption reduces both basal and Ang II-stimulated aldosterone production (**Figure 8D**).

4. ***“A burst has been defined for the examples of 300pM AngII. It would be important to show whether the same applies for other concentration of AngII.”***

Values were provided in the supplement. Table “Thresholds” in panel c of Figure S1 presents burst thresholds for each Ang II concentration.

5. ***“Line 122: not sure, why it is referred to Figure 2B.”***

Thanks for catching the error. Should be **Figure 3B**, line 129

Reviewer 2:

“The authors examine functional relationships between calcium activity of adrenal rosette cells. They find that cells display bursting behavior with a frequency that increases with the dose of Ang II. Further, the pattern of the calcium bursts can be clustered into functional groups with similar bursting patterns that are composed of spatially localized cells. I find the manuscript to be well written, the analysis appropriate, the results novel and interesting, and I have only the following minor comments and suggestions for improvement.”

Thank you for your kind comments and review.

1. ***“When performing the spatial analysis of functional clusters, the authors measure the average pairwise distance between cells within the same cluster and compare it to the pairwise distance***

between cells in different clusters. While this clearly shows that the functional clusters are spatially localized, it does not necessarily show that they are within the same rosette. For example, one functional cluster could span two rosettes that are located next to each other. I understand that it is difficult to visually identify the rosettes, but it would greatly strengthen the findings if the authors could find more evidence to convince the reader that the functional clusters showed high spatial overlap with rosettes. Potentially the authors could ask an expert blind to the functional findings to partition the image into rosettes, and an overlap measure could be calculated? Even if the partitioning wasn't exact, a high overlap would help to show that it is likely that the functional clusters do correspond to the spatial rosettes. The authors could also calculate the spatial extent (area defined by the convex hull) of the functional clusters and show that this corresponds with the expected spatial extent of a rosette, although this would not prove overlap."

We agree that the mean distance between cells within a functional cluster certainly does not address whether all cluster members reside in the same anatomical rosette. To provide a spatial representation of activity coordination and rosette mapping we have implemented a phase analysis (please see **response #3B to Reviewer #1**).

Phase analysis results can be depicted in a matrix plot. In phase analysis, the standard deviation of all spike latencies between two cells is calculated, and then all pair-wise comparisons are presented in matrix plot format. To quickly evaluate phase-locking among cells within the same rosette, the axes of the matrix plot are intentionally ordered according to rosette assignment; cells were assigned to rosettes by a blinded individual. Representative matrix plots (50pM, 300pM, 3nM Ang II, 1uM, **Figure 6**) show that cell-pairs with small activity standard deviations (blue-green hued squares) are more likely to be found within the same anatomical rosette, while cell-pairs with larger activity standard deviations (yellow-red squares) are more likely to be located in different rosettes.

- 2. "Although this is just terminology, I found it very confusing that the phrases "clustered pairs" and "non-clustered pairs" were used to describe the calculation of distances between cells. The functional clustering algorithm can lead to cells that are assigned to their own cluster (ie, individual cells that are not grouped with other cells – gray cells in Fig 6) which is what the phrase "non-clustered cells" brings to mind. Perhaps the phrases "within cluster pairs" and "between cluster pairs" or something similar can be used to avoid this confusion. Although the authors describe the meaning in the text, simply reading the figure without the corresponding text can be misleading.*

I have a similar comment regarding "centroid distance". It is easy to think that this relates to the centroid of the functional cluster, not of individual cells. I think something like "pairwise distance" would be less confusing."

We acknowledge that our original terminology was confusing and now use the suggested terminology: *within clustered-pairs*, *between clustered-pairs*, and *pairwise distance*.

- 3. "It would be useful to add a few words in line 138 to very briefly describe what the alternative thresholding method is so that the reader doesn't have to reference the SI just to have an idea of what the alternative was."*

Thanks for the suggestion. We now include a short description of the alternative thresholding method found on **page 7, lines 145-147**.

Reviewer 3:

“Guagliardo et al. studied Ca²⁺ signalling underlying angiotensin II (Ang II) -stimulated aldosterone production from rosettes of adrenal zona glomerulosa (zG) cells. Whereas isolated zG cells are known to respond to Ang II with little Ca²⁺ signalling, presently studied zG cells within rosettes expressing the Ca²⁺ indicator GCaMP3 showed highly reproducible and stereotypic Ca²⁺ responses to a wide range of Ang II concentration. This signalling was characterized by bursts of short Ca²⁺ spikes with a typical inter-spike period of about 2 sec. The bursts were separated by >5 sec periods with no or few irregularly occurring spikes. This Ca²⁺ activity depended on voltage-regulated Ca²⁺ channels and was prevented by a T-type channel blocker but was insensitive to disruption of Ca²⁺ release from the ER by the SERCA inhibitor cyclopiazonic acid. Unlike Ca²⁺ signalling in some other types of cells the inter-spike period was unaffected by the agonist concentration and the number of spikes per burst as well as burst duration were also concentration-independent. However, the lag-time between Ang II addition and the initial Ca²⁺ response decreased with increasing concentrations together with shortening of the inter-burst period resulting in increasing number of bursts with Ang II concentration.

Functional cluster analysis of Ca²⁺ spike synchrony between cell pairs revealed that closely located zG cells that likely belong to the same rosette formation show synchronized Ca²⁺ responses to Ang II that differ from the synchronized responses in other rosettes. Since zG cells lack synchronizing gap junctions and are not known to release factors that might mediate cell synchronization the authors focussed on other possible mechanisms. As N-cadherins (NCADs) are abundant and required for rosette formation, their possible importance for Ca²⁺ synchronization was investigated. A protocol with Ca²⁺ reduction to disrupt NCAD connections was followed by exposure to antibodies against the extracellular domain of NCAD to prevent reconnection upon subsequent restoration of the Ca²⁺ concentration. This approach resulted in somewhat reduced number of Ca²⁺ spikes in response to Ang II, whereas there was no effect when an antibody against the inaccessible intracellular NCAD domain was used. Interestingly, detailed analysis revealed that this effect was solely due to reduction of burst duration, with no effect on intraburst Ca²⁺ spike frequency or lag-time to the first burst. This effect contrasts to the shortening of lag time that was observed with increasing concentrations of Ang II.

The authors conclude that rosette-intrinsic NCAD coupling somehow ensures that Ang II can initiate coordinated Ca²⁺ spiking in zG rosettes to stimulate aldosterone production.

This interesting study is well written and I like the meticulous analysis of the signalling patterns. However, why reproducible Ca²⁺ spike patterns are generated only in zG cells within rosettes, occur in bursts and become synchronized among the rosette cells remains enigmatic.”

Thank you for acknowledging our comprehensive approach and detailed quantification of signaling and for providing your probing synopsis.

- 1. “Since not only T-type but also L-type Ca²⁺ channels have been implicated in the regulation of aldosterone production the authors should investigate how blockers of these channels affect Ang II-induced Ca²⁺ signalling in zG rosette cells.”***

We too are proponents of the importance of L-type Ca^{2+} currents in the regulation of aldosterone production. However, functional L-type channels in mouse (C57BL/6J) adrenal slices are not evident. Previously, we reported that neither the L-type channel antagonist nifedipine (1 μM), nor the L-type channel agonist BAYK8644 (1 μM), altered Ca^{2+} currents recorded from zG cells in mouse adrenal slices (Hu et.al. JCI 122:2046-2053, 2012). Nifedipine also failed to stop voltage oscillations. In contrast, oscillations were abolished by T-type Ca^{2+} channel blockers.

Notably, Cav1.2 and Cav1.3 channel mRNA is detected in micro-dissected zG layer tissue. These results, when viewed with our previous functional experiments, indicate that either L-type channel protein is not expressed, is not functional, or the detected mRNA is endothelial in this highly vascularized layer. Nevertheless, so that the observations in this manuscript can stand alone, we now include data showing that in contrast to TTA-P2, nifedipine does not inhibit Ang II-induced calcium oscillations (**Figure 2E**).

2. ***“It is interesting that Ang II concentration does not affect spike frequency but increases the number of bursts. No attempt is made to characterize the underlying ionic events in the bursting phenomenon.”***

We originally hoped to include such data in a separate study. While we still believe that a full characterization of the underlying conductances is beyond the scope of this manuscript (as it would require electrophysiological as well as imaging studies), we offer some new tantalizing data. We show that iberiotoxin, a blocker of Ca^{2+} -activated BK channels, and apamin, a blocker of Ca^{2+} -activated SK channels, alters zG Ca^{2+} oscillations (**Figure 2F and Supplemental Figure 2A**, respectively). We focused on interrogating Ca^{2+} -activated potassium conductances because we previously showed that blocking T-type Ca^{2+} channels halted voltage oscillations at the relatively depolarized membrane potential of -60mV, suggesting that reducing calcium influx also reduced the activity of hyperpolarizing potassium conductances.

3. ***“Although I like the analysis of signalling patterns I lack information of how well the Ca^{2+} spikes are synchronized between cells. The raster plots do not provide sufficiently detailed information to the reader, and it would be valuable with some discussion about the nature of the synchronization mechanism based on the relationship between individual spikes in different zG cells within a rosette. Does the coupling mechanism provide a basis for the simultaneous occurrence of spikes at random in different rosette cells or are the individual spikes somehow coupled between cells (occur with a certain time difference and/or order)?”***

We agree that the rasters are insufficient to visually resolve individual spiking events, and we regret not providing an adequate explanation of synchrony. Please see **response #3B to Reviewer #1** for a full explanation. Also, we now expand our discussion of synchrony in the manuscript contrasting it with phase-locking (see **page 9 lines 194-202 and page 9-11 lines 206-237**). In brief, we now include an additional, independent evaluation of the temporal relationships among active cells. Importantly, this approach also enables us to present such relationships in a more unambiguous manner. We now apply both approaches – functional clustering and phase analysis – to our experiments (**Figures 5 and 6**).

As indicated in our response to reviewer 2, cadherin disruption reduces the number of cells that functionally cluster, highlighting an important role for cadherins not only in determining the level of

activity but also the extent of synchrony. Importantly, our phase analyses show that phase-locked activity between cells is more likely to occur between cells within a rosette than between cells in different rosettes. Below, we have copied our response to **Reviewer #2 (Response #1)** for your convenience. Therein, we describe how this analysis determines fixed relationships and how we map the measure to anatomical rosettes:

“We agree that the mean distance between cells within a functional cluster certainly does not address whether all cluster members reside in the same anatomical rosette. To provide a spatial representation of activity coordination and rosette mapping that is not afforded by the functional clustering algorithm we have implemented a phase analysis” (please see response #3B to Reviewer #1).

Phase analysis results can be depicted in a matrix plot. In phase analysis, the standard deviation of all spike latencies between two cells is calculated, and then all pairwise comparisons are presented in matrix plot format. To quickly evaluate phase-locking among cells within the same rosette, the axes of the matrix plot are intentionally ordered according to rosette assignment; cells were assigned to rosettes by a blinded individual. Representative matrix plots (50pM, 300pM, 3nM Ang II, 1uM, **Figure 6**) show that cell-pairs with small activity variances (blue-green hued squares) are more likely to be found within the same anatomical rosette, while cell-pairs with larger activity variances (yellow-red squares) are more likely to be located in different rosettes.”

4. ***“I am not happy with the statement in lines 267-268 that “cellular junctions between closely apposed zG cells regulate zG calcium bursts”. In my view the word “regulate” is inappropriate since no regulatory role is demonstrated and the junctions may merely provide sufficient cellular compactness for another synchronizing mechanism.”***

We agree that cellular junctions may not be *the proximate regulator* of oscillatory activity and synchronization, and that cellular compactness and/or mechanosensitive conductances may play additional, significant roles. However, junctional strength is not fixed, but dynamic. The strength depends on EC-domain interactions that are Ca²⁺-dependent, and IC-domain interactions with the cytoskeleton that are continuously remade. Our data indicate that junctions are at least a part of the cascade. We have altered the text to reflect a compromise. **Line 307** regulate has been changed to support.

5. ***“In the same context I find the hypo-osmotic challenge data presented in Supplemental Fig. 3 interesting but difficult to interpret. Since the study deals with Ang II-induced Ca²⁺ responses it is confusing to discuss activation of the AT1-receptor in the absence of ligand.”***

We found these data interesting as well but have decided to remove it from the manuscript because we have now added significant quantities of new data.

6. ***“The effect of the calcium switch assay to disrupt cellular junctions is rather modest. Is it possible to somehow quantify the effectiveness of the cell junction disruption efficiency and relate it to the effect on Ca²⁺ signalling?”***

We agree that the disruption afforded by N-cadherin antibodies was modest and would need some measure of quantification to strengthen the story. Rather than go that route, we have modified our disruption protocol to include both anti-N and anti-K cadherin antibodies based on the

immunological data provided by our coinvestigator showing the presence of both cadherins in adrenal rosettes. These data are presented in a companion manuscript under review at *Nature Communications*. We now find a substantive reduction in oscillatory events that is quantified by a *significant decrease* in fractional active-time (**Figure 7C, D insets**). The fewer events observed in N/K-cadherin antibodies also results in both a significant decrease in burst number and a significant shortening of burst duration. We provide event disruption data at two hormone concentration of Ang II (300pM and 3nM). In addition, we show that N/K antibodies decrease baseline and Ang II-stimulated aldosterone production.

7. Add time scale to Fig. 1B

Corrected. Thanks for identifying all minor issues.

8. Clarity in Fig. 2E would be improved by green symbols indicating “3 nM Ang II + 20 μM CPA” rather than “+20 μM CPA”.

Corrected.

9. Line 257, with reference to Fig S4 I suppose 300nM is a misprint and should be 300pM.

Corrected.

10. Panel C in Fig 5 indicates 300 nM Ang II but the figure legend states 300 pM.

Corrected. Now figure S5a

11. The X-axis in Fig. 6B is unlabelled.”

Corrected.

REVIEWERS' COMMENTS:

Reviewer #1 (Remarks to the Author):

The authors have provided additional data that answer my questions. Overall, this has further strengthened the case.

Minor remark:

I suggest to adjust the x-axis of Figure 3e and 3f using the same scale

Reviewer #2 (Remarks to the Author):

The authors have addressed my concerns and I believe the paper is ready for publication.

Reviewer #3 (Remarks to the Author):

The study by Guagliardo et al. was further improved and strengthened by the revision and represents an inspiring piece of work. Like before I am particularly impressed by the analysis. The new data on aldosterone release and N/K-Cad antibody disruption are valuable but the main weakness of the study remains, the lack of identification of the synchronizing mechanism. I have some additional specific comments to consider.

1. In lines 107-116 the authors state that the more rapid onset of evoked oscillations in the presence of the K⁺ channel blocker iberiotoxin (cited data) presumably reflects decreased oscillatory threshold by a depolarizing shift in the zG cell membrane potential. Later in the same section the observed lack of influence of CPA is taken to indicate that the Ca²⁺ events are not dependent on ER calcium stores. Another aspect in this context is that CPA not only empties Ca²⁺ from the ER but also activates a depolarizing store-operated cation influx and nevertheless fails to affect the response. This discrepancy requires a comment.

2. Line 298 refers to Fig 7e f, however, panel "f" in Fig 7 is mislabelled as "g". Moreover, I think that the Ang II concentration in this panel is 3 nM rather than 300 pM as stated in the legend to Fig 7 (line 860).

Resubmission Rebuttal

Reviewer #1 (Remarks to the Author):

- The authors have provided additional data that answer my questions. Overall, this has further strengthened the case.

Thank you.

- Minor remark:
I suggest to adjust the x-axis of Figure 3e and 3f using the same scale

Done.

Reviewer #2 (Remarks to the Author):

- The authors have addressed my concerns and I believe the paper is ready for publication.
Thank you.

Reviewer #3 (Remarks to the Author):

- The study by Guagliardo et al. was further improved and strengthened by the revision and represents an inspiring piece of work. Like before I am particularly impressed by the analysis. The new data on aldosterone release and N/K-Cad antibody disruption are valuable but the main weakness of the study remains, the lack of identification of the synchronizing mechanism. I have some additional specific comments to consider.

Thank you.

- In lines 107-116 the authors state that the more rapid onset of evoked oscillations in the presence of the K⁺ channel blocker iberiotoxin (cited data) presumably reflects decreased oscillatory threshold by a depolarizing shift in the zG cell membrane potential. Later in the same section the observed lack of influence of CPA is taken to indicate that the Ca²⁺ events are not dependent on ER calcium stores. Another aspect in this context is that CPA not only empties Ca²⁺ from the ER but also activates a depolarizing store-operated cation influx and nevertheless fails to affect the response. This discrepancy requires a comment.

Your point is well taken. By depleting calcium stores, CPA can activate STIM/Orai channels, thereby introducing a small inward current within cells. The contribution of this conductance to zG cell membrane potential remains unknown, but is presumed to be minimal because the single channel conductance of ORAI channels is extremely small (10-25 fS, reviewed in Prakriya & Lewis, 2015). As we saw no effect with CPA, we presume that the contribution of ORAI channels to zG cell calcium oscillations is negligible. We now describe this possibility on line 112-114 of the manuscript.

- Line 298 refers to Fig 7e f, however, panel “f” in Fig 7 is mislabelled as “g”. Moreover, I think that the Ang II concentration in this panel is 3 nM rather than 300 pM as stated in the legend to Fig 7 (line 860).

Thanks for finding our error. We have now fixed the figure and clarified the legend.